# Tuning hydrogenation chemistry of Pd-based heterogeneous catalysts by introducing homogeneous-like ligands

Jianghao Zhang [1,2,6], Wenda Hu[1,3,6], Binbin Qian[4,5], Houqian Li[1], Berlin Sudduth[1], Mark Engelhard [3], Lian Zhang [5], Jianzhi Hu [1,3], Junming Sun[1] ✉, Changbin Zhang [2] ✉, Hong He [2] & Yong Wang [1,3] ✉

Noble metals have been extensively employed in a variety of hydrotreating catalyst systems for their featured functionality of hydrogen activation but may also bring side reactions such as undesired deep hydrogenation. It is crucial to develop a viable approach to selectively inhibit side reactions while preserving beneficial functionalities. Herein, we present modifying Pd with alkenyl-type ligands that forms homogeneous-like Pd-alkene metallacycle structure on the heterogeneous Pd catalyst to achieve the selective hydrogenolysis and hydrogenation. Particularly, a doped alkenyl-type carbon ligand on Pd-Fe catalyst is demonstrated to donate electrons to Pd, creating an electron-rich environment that elongates the distance and weakens the electronic interaction between Pd and unsaturated C of the reactants/products to control the hydrogenation chemistry. Moreover, high $H_2$ activation capability is maintained over Pd and the activated H is transferred to Fe to facilitate C-O bond cleavage or directly participate in the reaction on Pd. The modified Pd-Fe catalyst displays comparable C-O bond cleavage rate but much higher selectivity (>90%) than the bare Pd-Fe (<50%) in hydrotreating of diphenyl ether (DPE, modelling the strongest C-O linkage in lignin) and enhanced ethene selectivity (>90%) in acetylene hydrogenation. This work sheds light on the controlled synthesis of selective hydrotreating catalysts via mimicking homogeneous analogues.

The remarkable hydrogen activation of noble metals (e.g., Pd) has found wide applications in catalytic hydrotreating reactions such as hydrogenolysis of C–O bonds in lignocellulosic biomass[1,2] and hydrogenation of specific functional groups[3–5]. However, noble metals also strongly interact with sp² carbon in C=C bond and aromatic ring[6,7], leading to undesirable deep hydrogenation, e.g., aromatic ring hydrogenation along with C–O bond scission in the lignin valorization process or total hydrogenation of all the unsaturated bonds during hydrotreating of alkyne/alkene feedstock in the alkene polymerization industry. Those undesirable reactions not only dramatically reduce carbon efficiency but consume excess hydrogen[8]. It is still highly challenging to control the selective hydrogenation chemistry of noble-metal-based catalysts.

In homogeneous catalysis, coordinative ligands have been widely studied to exquisitely tailor the electronic property of metal centers or

[1]The Gene & Linda Voiland School of Chemical Engineering and Bioengineering, Washington State University, Pullman, WA 99164, USA. [2]State Key Joint Laboratory of Environment Simulation and Pollution Control, Research Center for Eco-environmental Sciences, Chinese Academy of Sciences, Beijing 100085, China. [3]Institute for Integrated Catalysis, Pacific Northwest National Laboratory, Richland, WA 99352, USA. [4]School of Chemistry and Environmental Engineering, Yancheng Teachers University, Yancheng 224002, China. [5]Department of Chemical Engineering, Monash University, Clayton, Victoria 3800, Australia. [6]These authors contributed equally: Jianghao Zhang, Wenda Hu. ✉e-mail: junming.sun@wsu.edu; cbzhang@rcees.ac.cn; yong.wang@pnnl.gov

geometric structure of organometallics to achieve high selectivity toward desirable products[9]. For instance, the selectivity of semi-hydrogenation of alkyne to alkene over Pd complex was significantly enhanced (from <60% to >95%) after being coordinated with an optimized carbene ligand[10]. The selective dehydrogenation of alkane to alkene could be accomplished over homogeneous metallacycle bearing platinum-group metals[11]. In heterogeneous catalysis, while catalyst deactivation by carbonaceous deposition still prevails, the benefits of carbon decoration on the performance of catalysts have been recently demonstrated[3,12–14]. Subsurface carbon in the lattice of Pd was found to significantly reduce the population of non-selective subsurface hydrogen, enhancing the selective hydrogenation of alkyne to alkene[3,15] (similar performance as that of homogeneous carbene-coordinated Pd[10]). In addition, short-range chains of carbon oligomers over Pt black were reported to inhibit skeletal reaction (i.e., isomerization and cyclization) of n-hexane, promoting dehydrogenation products (i.e., hexenes)[16] (similar performance as that of homogeneous metallacycle bearing platinum-group metals[11]). Though the mechanisms in these catalytic processes have not been comprehensively understood, the selective catalysis upon carbon decoration or ligand coordination in hetero and homogeneous hydrogenation/dehydrogenation reactions suggests a similar ability to tailor the surface chemistry of noble metal catalysts.

Previously, we developed a bifunctional catalyst employing Fe for C−O bond activation and Pd for reductive elimination of formed radicals from C−O cleavage[17,18]. While catalyst reactivity increases with Pd loading, the increased Pd loading was found to aggravate deep hydrogenation. In this work, we report a gas-phase deposition method, by which a uniquely stable carbonaceous species can be generated on the Pd nanoparticles. More importantly, the presence of such carbonaceous species significantly improved the selectivity of C−O bond cleavage during hydrotreating of diphenyl ether (modeling the 4-O-5 linkage in lignin) and semi-hydrogenation of acetylene. Complementary characterization techniques including high-angle annular dark field scanning transmission electron microscopy (HAADF-STEM) coupled with elemental mapping, near edge X-ray absorption fine structure (NEXAFS), magic angle spinning nuclear magnetic resonance (MAS NMR), temperature-programmed experiments, in situ X-ray photoelectron spectroscopy (XPS) and in situ attenuated total reflectance Fourier-transform infrared spectroscopy (ATR-FTIR), as well as density functional theory (DFT) modeling revealed that the Pd is coordinated with an alkenyl type of ligand forming homogeneous-like metallacycle structure. As a result, an electron-rich Pd was generated which weakens the electronic interaction between Pd and unsaturated $sp^2$ C, leading to inhibited hydrogenation of C=C bonds in the reactants/products.

## Results

### Performances of the catalysts

Using DPE as a probe molecule to mimic the strongest C−O linkage (i.e., 4-O-5 type) in lignin, we evaluated the catalytic performance of Pd-Fe catalysts in liquid-phase hydrotreating. Given the non-tautomerization nature of DPE[19], the primary products were only obtained from two reaction pathways: direct hydrogenation to dimers that are ring-saturated without C−O bond cleavage[19] and C−O bond cleavage to monomers, i.e., benzene and phenol (Supplementary Fig. 1 and Note 1). The produced phenol may be further converted to deoxygenated hydrocarbons. As suggested by the catalysts' performance as a function of time-on-stream (Supplementary Fig. 2), monometallic Fe led to exclusive C−O bond cleavage and monometallic Pd (i.e., Pd/C) directly hydrogenated the aromatic ring of DPE with ~92% selectivity. Therefore, the calculated turnover frequency (TOF) of C−O bond cleavage and direct hydrogenation were based on exposed Fe and Pd, respectively (details are shown in Supplementary Method, Supplementary Fig. 3 and Supplementary Table 1). Figure 1a

displays the reactivities of C−O bond cleavage over different catalysts. Compared to the monometallic Fe, addition of Pd (XRD patterns are shown in Supplementary Fig. 4) enhanced the activity of C−O bond cleavage and increasing the Pd weight loading to 5% (denoted as 5Pd/Fe) promoted the TOF by ~7 fold. However, increased Pd loading also led to a dramatically increased reactivity in direct hydrogenation (Fig. 1b), and thus decreased selectivity to C−O bond cleavage (Fig. 1c). This result is presumably attributed to the formation of larger Pd particle that regains the nature of monometallic Pd[20] for direct hydrogenation of the aromatic ring.

Interestingly, after a pretreatment of 5Pd/Fe with 0.4 vol% CO/$H_2$, while the C−O bond cleavage activity is maintained, the direct hydrogenation activity was dramatically inhibited on the 5Pd-C/Fe-C (Fig. 1a, b and Supplementary Fig. 5), compared with the 5Pd/Fe. In particular, the 5Pd-C/Fe-C catalyst exhibited >90% selectivity towards C−O bond cleavage which is significantly higher than that of 5Pd/Fe (Fig. 1c and Supplementary Fig. 6). Note that our separate control synthesis revealed that the selective removal of carbonaceous species from Pd (i.e., 5Pd/Fe-C) revived the direct hydrogenation activity (Fig. 1b). This result suggests the importance of carbonaceous species on the Pd nanoparticles, which will be further discussed in the following sections. Durability tests of 5Pd-C/Fe-C revealed that both the reactivity and high C−O bond cleavage selectivity remained during 6 testing cycles (Supplementary Fig. 7), indicating that the carbon species are stable under the reaction conditions. The carbon species are not likely carbidic carbon since the carbidic one could be removed from Pd in $H_2$ above 150 °C[21], and the harsher conditions of the hydrotreating reaction would exclude the carbidic carbon as an effective modifier. In addition, when a higher CO concentration (i.e., 4%CO/$H_2$) was used during the synthesis to generate thin graphitic overlayers on Pd nanoparticles[12], severe catalyst deactivation was observed (Supplementary Fig. 8 and Note 2). This result suggests that surface graphene overlayers do not likely cause the enhanced selectivity in 5Pd-C/Fe-C.

The 5Pd-C/Fe-C catalyst was also investigated for hydrogenation of acetylene (possible reaction pathways are shown in Supplementary Fig. 9). At comparable conversions, both 5Pd/Fe and 5Pd-C/Fe-C show a short induction period in selectivity at the beginning of the reaction (Fig. 1d and Supplementary Fig. 10), which has been well known to correlate to the formation of subsurface carbon species[3]. After the induction period of about 20 min, ethene selectivity remains stable at >90% over 5Pd-C/Fe-C as opposed to ~80% over 5Pd/Fe (Fig. 1d). This suggests that the deposited carbon species are distinct from the in situ formed subsurface carbon[3] during selective hydrogenation to ethene. In summary, the pretreatment of 5Pd/Fe catalyst generated surface carbon species on the Pd nanoparticles, which in turn preserved the beneficial functionality of Pd (i.e., the enhanced activation of $H_2$) but selectively inhibited the undesirable deep hydrogenation of C=C bond in the aromatics or ethylene.

### Structures of the catalysts

XAS was performed to unravel the structure of the catalysts. The X-ray absorption near edge structure (XANES) of Fe (Supplementary Fig. 11 and Note 3) for the three 5% Pd-Fe catalysts show the position of the absorption edge, the intensities of the pre-edge and white line features that resemble the standard Fe foil. In XANES of the Pd K edge (Supplementary Fig. 12 and Note 3), the samples also display spectra similar to that of Pd foil, indicating that Pd exists in the metallic state in all catalysts. The local atomic structure of Pd is investigated with Fourier-transformed extended X-ray absorption fine structure (EXAFS, Supplementary Fig. 13 and Note 3). The main peak shows up at ~2.4 Å that is attributed to Pd-Pd first shell[22]. Compared with Pd foil, spectra of all catalysts display a decay of the main peak, leading to the emergence of the abreast peak at ~2.2 Å. The similar change has been widely reported for the Pd-Fe bimetallic system and ascribed to the interference

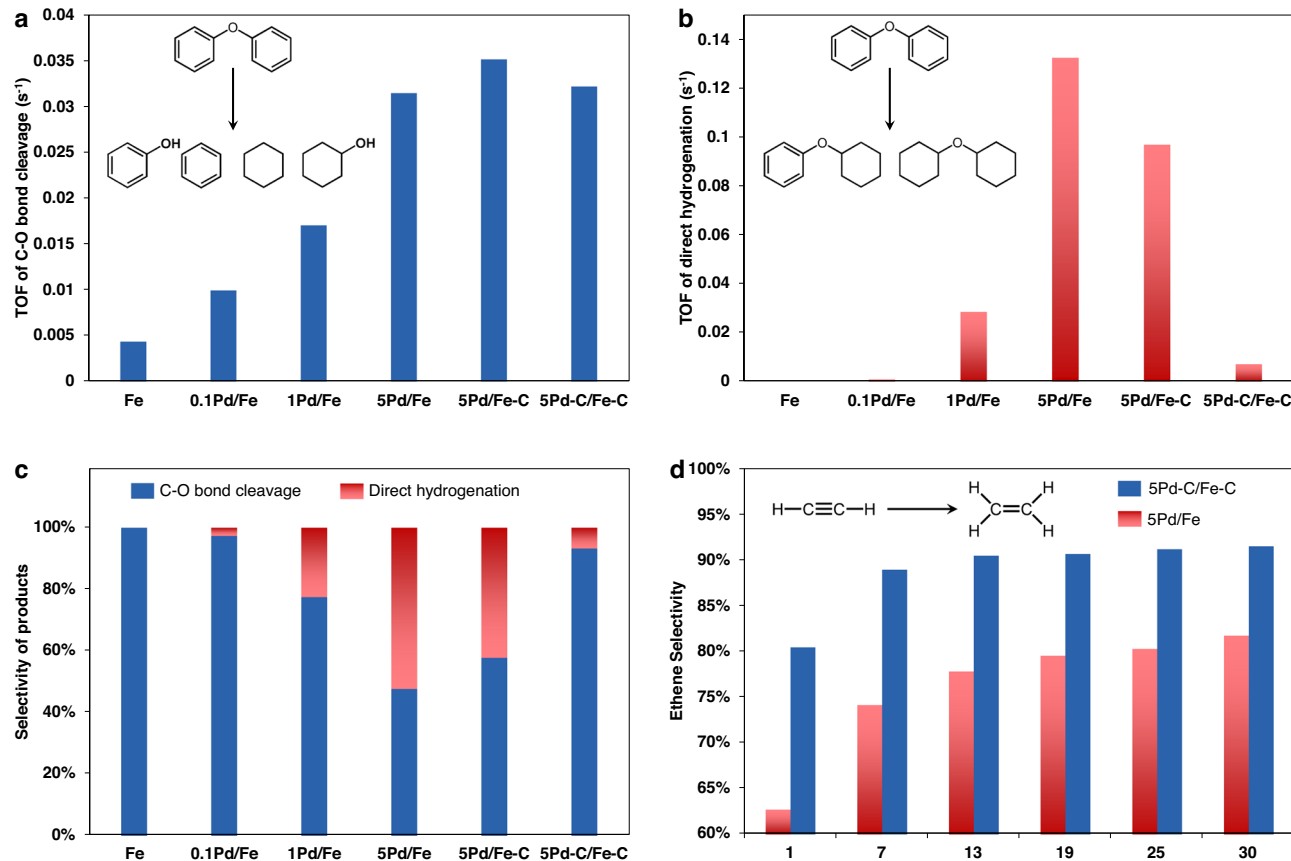

**Fig. 1 | Performance of Pd-Fe catalysts in selective hydrogenolysis of DPE and hydrogenation of acetylene.** The weight percentages of Pd in the catalysts are 0, 0.1, 1, and 5. **a**, **b** Turnover frequencies (TOFs) of C−O bond cleavage (**a**, based on exposed Fe) and direct hydrogenation (**b**, based on exposed Pd) over different catalysts (at <20% conversion). **c** Selectivity of different categories of products at -10% conversion (except that the selectivity for monometallic Fe was derived at -5% conversion). Reaction condition: 0.03 g catalyst, 50 mL $C_{16}H_{34}$, 1.08 g DPE, 250 °C, 5.6 MPa $H_2$. **d** Ethylene selectivity of catalysts for hydrogenation of acetylene. Reaction condition: 0.03 g catalyst, 4% $C_2H_2$−8% $H_2$ balanced with $N_2$, GHSV = 120,000 mL/(g*h), 150 °C, 0.1 MPa.

between Pd-Pd and Pd-Fe oscillations[23,24]. Structural parameters determined by EXAFS curve-fitting for the three catalysts are nearly identical (Supplementary Table 1), indicating comparable structure of Pd particles in all the catalysts. High-resolution transmission electron microscopy (HRTEM) was employed to study the 5Pd/Fe, 5Pd/Fe-C, and 5Pd-C/Fe-C catalysts. All the three samples display similar morphologies, i.e., small Pd particles on a Fe support (Fig. 2a). The measured interplanar spacing for the lattice fringes of the supported particles is -0.22 nm which corresponds to the (111) plane of face-centered cubic Pd[25]. The support has an interplanar distance of -0.21 nm, typically attributed to (110) plane of body-centered cubic Fe[26]. The particle sizes of Pd over 5Pd/Fe, 5Pd/Fe-C, and 5Pd-C/Fe-C were first determined via the CO chemisorption results (12–15 nm, Supplementary Table 1). The comparable sizes were further substantiated by a statistical analysis of microscopic images of these catalysts (Supplementary Fig. 14 and Supplementary Table 1). It should be noted that 5Pd-C/Fe-C has a larger deviation between particle sizes of Pd determined by CO pulse chemisorption and TEM, than the other two catalysts. This may be due to that the ligands on Pd occupy certain sites so that the particle size is overestimated by CO chemisorption method. Overall, these results suggest the Pd and Fe in the three catalysts share identical structures, i.e., metallic Pd nanoparticles of similar size supported on bulk metallic Fe.

The location of carbon species was analyzed with HAADF-STEM and EDS elemental mapping, as shown in Fig. 2b–d. Besides the Pd on Fe, the mapping of 5Pd/Fe detects only a small amount of C that is attributed to adventitious carbon[27]. While carbon is detected on both

Pd and Fe domains of 5Pd/Fe-C and 5Pd-C/Fe-C, the C signals are considerably lower on the Pd domain of the 5Pd/Fe-C catalyst than the 5Pd-C/Fe-C albeit slightly brighter than that of the Pd of 5Pd/Fe. This result suggests the synthesis protocol used did not completely, but instead partially removed the C deposited on Pd for the 5Pd/Fe-C samples, which is consistent with the observed slight inhibition for direct hydrogenation over 5Pd/Fe-C as compared to that of 5Pd/Fe (Fig. 1b, c). Based on the above results, structures of the three catalysts are proposed and schematically presented in Fig. 2e. Compared with the clean 5Pd/Fe and the 5Pd/Fe-C that show C on both Pd and Fe region, the Pd in 5Pd-C/Fe-C is covered with a significantly higher amount of C species. In addition, the fresh and spent catalysts were also analyzed with XRD. No changes corresponding to metallic Pd and Fe were observed for all the catalysts (Supplementary Figs. 15–16 and Note 4). Consistent with the XRD results, TEM images of all the spent catalysts show similar Pd particle size on the Fe substrate (Supplementary Fig. 17). Since the spent catalysts may adsorb carbon-containing reactants, products or solvents that can overlap with the signal from C deposit, elemental mapping of C was not performed.

Temperature-programmed reactions with $H_2$ ($H_2$-TPRea) of in situ synthesized catalysts were carried out to further study the carbon species, as evidenced by the gas-phase methane formation ($m/z = 15$ in Fig. 3a). 5Pd/Fe shows a small broad feature with noise that may originate from the adventitious carbon[27]. 5Pd/Fe-C displays a peak located at 439 °C that can be attributed to the gasification of carbon aggregates on Fe[28]. Over 5Pd-C/Fe-C, besides the peak at 439 °C, a new methane peak at 361 °C is observed. Separate $H_2$-TPRea on synthesized

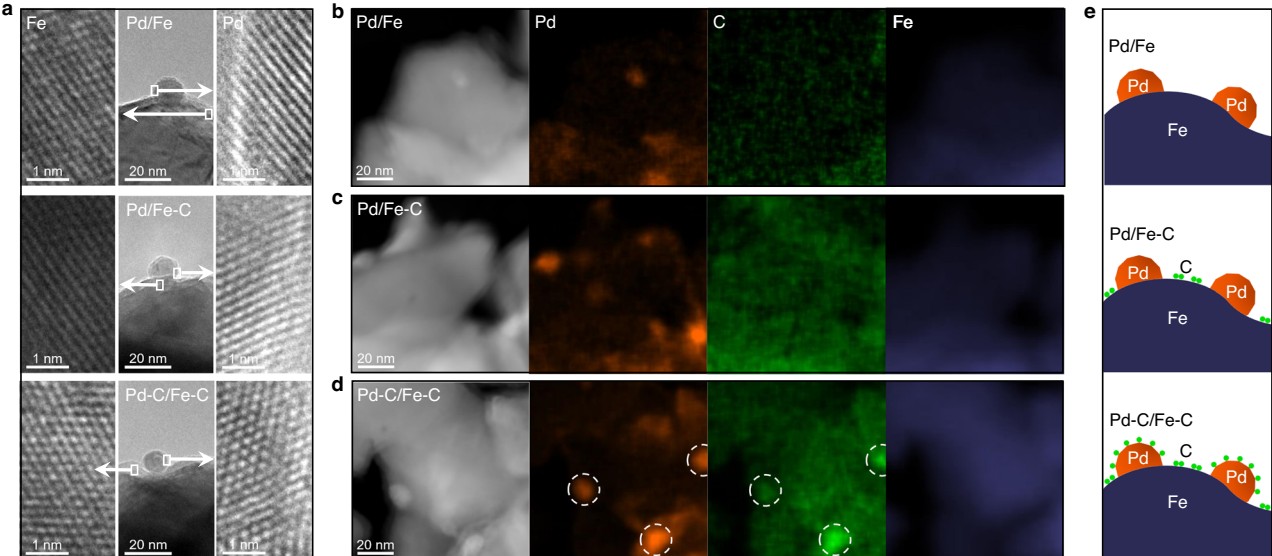

**Fig. 2 | Microscopic analysis of Pd/Fe, Pd/Fe-C, and Pd-C/Fe-C. a** HRTEM images of samples, the left and right ones show the regions in white squares; **b–d** STEM images and EDS elemental mapping of Pd, C, Fe over 5Pd/Fe (**b**), 5Pd/Fe-C (**c**), and 5Pd-C/Fe-C (**d**), Brown: Pd, green: C, navy: Fe; **e** schematic illustration of carbon locations on Pd and Fe. All samples were pretreated with 10%H$_2$/Ar at 250 °C for l h to simulate the catalysts after the hydrotreating reaction.

Pd-C/CNT (Supplementary Fig. 18 and Note 5) suggests this low-temperature peak should be attributed to the carbon species on Pd. The structure of doped carbon was further investigated with C 1s NEXAFS spectra (Fig. 3b). The peak near 285.0 eV represents π* resonance of sp$^2$ carbon species[29], which has slightly higher energy than that of graphite or aromatic species at 284.8 eV (graphite reference). The shoulder peak at 287.3 eV corresponds to the π* excitations of aliphatic C–H group[30]. The peak at higher energy of 288.3 eV is related to an electron-deficient carbon species with π*$_{C=O}$ resonance[30], which may be contaminant in this measurement, since this type of carbon was not detected in the XPS of pseudo in situ pretreated catalysts (Supplementary Fig. 19). In the δ* region, the peak at 292.8 eV is typically attributed to the C–C resonance[31]. One detectable difference of the spectra is that 5Pd-C/Fe-C has a higher intensity for peak associated with sp$^2$ carbon at ~285.0 eV, as shown in the inset of Fig. 3b. It indicates that the carbon species generated on Pd has an alkenyl structure. Using Fe for charge correction (Supplementary Fig. 20) and 5Pd/Fe as background, C 1s XPS difference spectrum of 5Pd-C/Fe-C displays a peak located at 284.1 eV that has been attributed to a chain hydrocarbon bonded to metal (Supplementary Fig. 19 and Note 6)[16], as opposed to the carbide (282.9–283.4 eV)[32,33], graphitic layer (284.5–284.8 eV)[16,34], or oxygenates (286.0–289.0 eV)[34]. In the Pd 3d region (Fig. 3c), the Pd 3d$_{5/2}$ peak of 5Pd/Fe appears at 336.2 eV which is typically reported for Pd-Fe bimetallic materials[18,35]. The peak for 5Pd-C/Fe-C shifts to a lower binding energy (336.0 eV), suggesting the alkenyl carbon species donate electrons to Pd. This is consistent with the first derivative of XANES (Supplementary Fig. 21) showing that the energy of absorption edge ($E_0$) of 5Pd-C/Fe-C (24,349.6 eV) is slightly lower than those of Pd foil (24,350.0 eV), 5Pd/Fe (24,349.9 eV), and 5Pd/Fe-C (24,349.8 eV), which indicates the Pd species over 5Pd-C/Fe-C is more electron-rich[36]. Besides the direct interaction between Pd and carbon species, one possible mechanism of electronic modification of Pd is that carbon species affects the Pd-Fe interaction. However, the EXAFS results showed no obvious change of Pd-Fe coordination number by a 0.4 vol% CO/H$_2$ treatment (Supplementary Fig. 13 and Table 2). More importantly, no selective chemistry was observed on 5Pd/Fe, suggesting the Pd-Fe alloy effect should be excluded. In addition, the Fe-free catalysts (i.e., CNT-supported ones) also show improved selective hydrogenation chemistry as the Fe-supported ones after the similar treatment with 0.4 vol% CO/H$_2$ (Supplementary

Figs. 22 and 23). Those results suggest that Fe-Pd alloy or Fe modified Pd should not play a major role in regulating the catalytic performance.

To further validate the structure of the carbonaceous species on the Pd nanoparticles, carbon deposition was done on Pd using $^{13}$CO as carbon source to amplify the $^{13}$C abundance so that the quantity of spin nuclei was higher than the detection limit of NMR. Solid-state $^{13}$C NMR was then employed to examine the chemical structure of this $^{13}$C-labeled species (Fig. 3d). To make the NMR study operable, the magnetic Fe support was replaced by carbon nanotube since the tailored hydrogenation chemistry mainly occurs on Pd. Indeed, the selective hydrogenation chemistry of $^{13}$CO-treated Pd/CNT was verified by the inhibition of ring saturation during DPE hydrotreating (Supplementary Fig. 22). Selective hydrogenation of acetylene to produce ethylene was also confirmed on the Pd-C/CNT catalysts (Supplementary Fig. 23). A comparison of the catalytic performances of Pd-C/CNT and 5Pd-C/Fe-C for acetylene hydrogenation is shown in Supplementary Fig. 24. Compared with the Pd/CNT reference, Pd-$^{13}$C/CNT displays two additional peaks located at 62 and 152 ppm that can be attributed to sp$^3$ and sp$^2$-hybridized carbon[37], respectively. Note that NMR shielding is significantly influenced by the coordinating atoms, and simply referring to NMR archives of carbon molecules without a Pd substrate may not pinpoint the structure of doped carbon species. Therefore, NMR shielding and corresponding chemical shift of possible carbon structures were then calculated by the gauge invariant atomic orbital (GIAO) method[38,39]. A cluster model, rather than a larger planar facet, was used as the Pd substrate since effective carbon species could not be formed in the latter case (Supplementary Fig. 25 and Note 5). Our activity tests have excluded carbidic, subsurface, or graphitic carbon as effective carbon species in 5Pd-C/Fe-C (vide supra). Among other examined structures, i.e., alkene-type palladacycle[40], alkane-type palladacycle (linear δ bonding)[41], metallocene analog[42], alkene on Pd with δ-bonding[43], alkane-type carbon coordinating Pd with bridge configuration[44] and carbene-like structure[10] (details are illustrated in Supplementary Fig. 26 and Note 7), alkene-type palladacycle has the closest NMR shifts (Fig. 3e) when compared to the experimental results (Fig. 3d). Based on the above results, it is inferred that the homogeneous-like alkenyl-type carbon in palladacycle and its analogs play a pivotal role in tailoring hydrotreating chemistry. The alkenyl-type carbon moiety may form via carbide intermediate at the low synthesis temperature[45], while its overgrowth, which may form

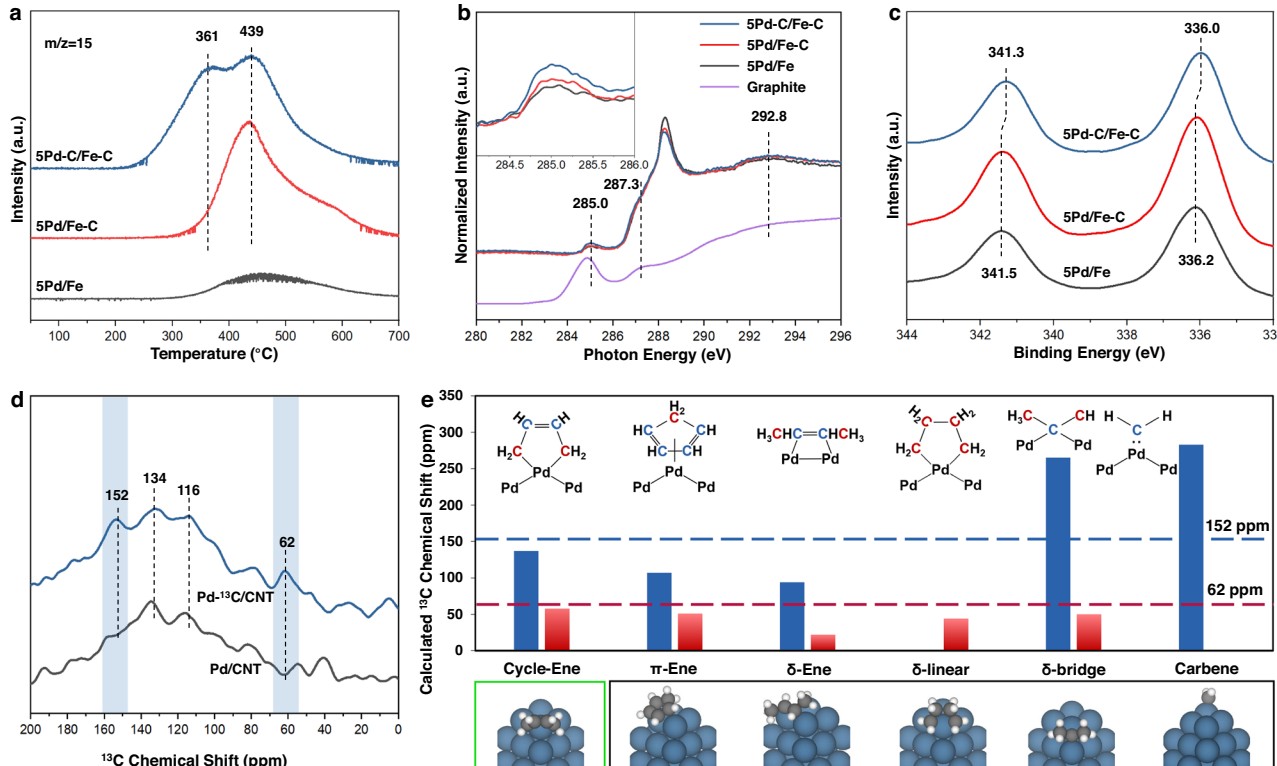

**Fig. 3 | Investigation of the carbon species and the electronic modification towards Pd. a** Methane formation ($m/z = 15$) profiles during $H_2$-TPRea of the catalysts; **b** normalized NEXAFS spectra at C 1$s$ K-edge. The spectra were collected in total electron yield (TEY) mode; **c** Pd 3d region of pseudo-in situ XPS of catalysts pretreated with 10%$H_2$/Ar at 250 °C for 0.5 h; **d** $^{13}$C MAS solid-state NMR of modeling samples. The $^{13}$C was doped on Pd with the same protocol as 5Pd-C/Fe-C except for using $^{13}$CO as carbon source. **e** Predicted $^{13}$C NMR chemical shifts of different carbon species on Pd cluster. Cycle-Ene: palladacycle with alkenyl-type carbon; π-Ene: metallocene; δ-Ene: alkene on Pd with δ-bonding; δ-linear: alkane-type palladacycle with linear δ bonding configuration; δ-bridge: alkane on Pd with bridge configuration; Carbene: carbene-like ligand on Pd. The red or blue bars, respectively, correspond to the C atoms labeled with same color.

graphitic layer that deactivates the Pd (Supplementary Fig. 8), is inhibited by the high concentration of $H_2$.

## Investigation of the working mechanisms

In situ ATR-FTIR was employed to investigate the interaction between DPE and the catalyst surface. For comparison purpose, free DPE with the absence of catalyst was also studied with ATR-FTIR (Supplementary Fig. 27 and Note 8). The asymmetric $v8$ band was deconvoluted as $v8a$ at 1591 cm$^{-1}$ and $v8b$ splitting modes at 1582 cm$^{-1}$ (Wilson's numbering[46,47]) with the latter one being much more intense. The $v8b$ vibration mode originates from the electron resonance between the aromatic ring with the substituent[48] (i.e., phenoxy group in this particular case). It is reported that the $v8$ band has high sensitivity to molecular perturbation such as ring deformation and symmetry losses[49]. Figure 4a shows the spectra in the C−C stretching and $C_{ring}$-H bending region recorded in the absence of $H_2$ cofeeding. All the samples display peaks at 1570, 1541, 1471, and 1456 cm$^{-1}$, which can be attributed to the vibrational mode of $v8a$, $v8b$, $v19a$, and $v19b$ + $C_{ring}$-H bending in the Wilson's numbering[46,47], respectively. Noteworthy, compared with the other two samples, the relative ratio of $v8b$ to $v8a$ on 5Pd-C/Fe-C is much higher, which is similar to that of free DPE. This observation suggests that the electron resonance in the aromatic ring on Pd is inhibited over 5Pd/Fe and 5Pd/Fe-C, whereas the electron resonance on 5Pd-C/Fe-C is less perturbed, likely due to weakened interaction between the aromatic ring and Pd surface caused by the deposited carbon species. The adsorption of DPE in the presence of $H_2$ was also investigated and shown in Fig. 4b. Compared with the spectra without $H_2$ cofeeding, the locations for absorption of carbon skeleton vibration remains unchanged, except that the peak at 1456 cm$^{-1}$ is

shifted to 1433 cm$^{-1}$. Given $v19a$ and $v19b$ are the degenerated pairs in vibration mode[50], the unchanged peak positions of skeleton vibration indicate that the shifted peak is attributed to perturbed $C_{ring}$-H bending. Since the bare Fe and Fe-C contribute no absorption at 1433 cm$^{-1}$, the peak at 1433 cm$^{-1}$ on all the Pd-Fe catalysts should originate from DPE adsorbed on Pd. Moreover, the intensity ratio of 1433 cm$^{-1}$ ($C_{ring}$-H bending) to 1471 cm$^{-1}$ ($v19a$) on 5Pd-C/Fe-C (0.79) is much lower than that on 5 Pd/Fe (1.37) and 5 Pd/Fe-C (1.48). Since absorption intensity increases with bond polarity[49], the lower ratio of 5Pd-C/Fe-C indicates $C_{ring}$-H bending becomes less polar, suggesting the adsorbed DPE has a similar state as free DPE (Supplementary Fig. 27). The similar polarity of $C_{ring}$-H in $H_2$ atmosphere, together with similar electron resonance of the aromatic ring in DPE on 5Pd-C/Fe-C, compared with the free DPE, both suggest that the coordinative carbon weakens the interaction between the aromatic ring and Pd. And this weakening effect is independent of the $H_2$ atmosphere, although the perturbation to the ring is more severe in $H_2$. The interaction between ethene and Pd was investigated by temperature-programmed desorption of ethene ($C_2H_4$-TPD, Fig. 4c). Each catalyst displays two desorption peaks that may be attributed to ethene adsorbed with different configurations on Pd (e.g., π-bonded and di-δ-bonded)[51,52]. The 5Pd/Fe and 5Pd/Fe-C samples show the desorption of ethene in the region of 0−80 °C, while the desorption temperature on 5Pd-C/Fe-C is 20 °C lower, implying a weaker interaction between the catalyst surface and ethene. It should be noted that the desorption peaks of 5Pd-C/Fe-C have comparable shape with the other two catalysts. It indicates that all the catalysts have similar sites for the adsorption of ethene, other than blocking of a specific kind of site by the coordinative carbon on 5Pd-C/Fe-C for the selective hydrogenation. Moreover, the coordinative carbon does not

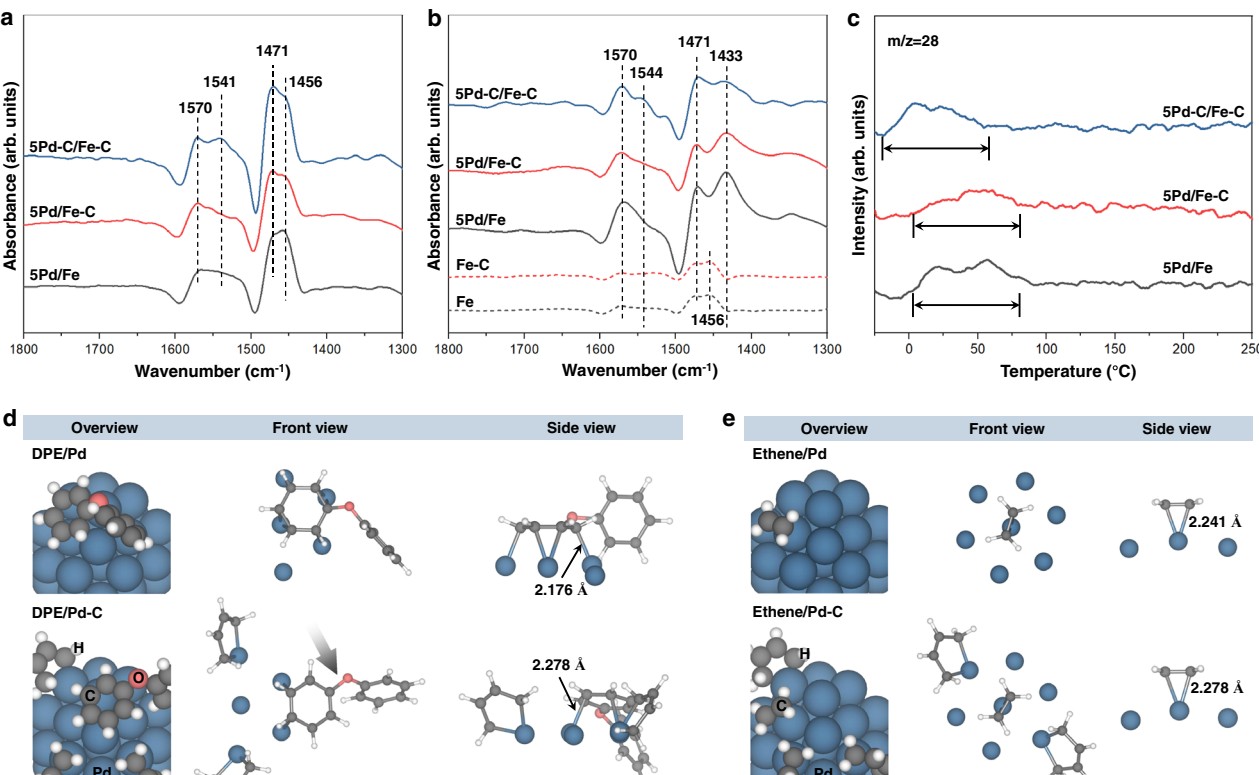

**Fig. 4 | Investigation of mechanisms in the selective hydrotreating. a, b** In situ ATR-FTIR spectra of DPE adsorption on Fe-based catalysts with absence (**a**) and presence (**b**) of H$_2$. **c** C$_2$H$_4$-TPD profiles of catalysts. **d** Optimized structure for modeling of DPE adsorbed on clean and carbon-modified Pd cluster. **e** Optimized structure for modeling of ethene adsorbed on clean and carbon-modified Pd cluster.

influence the activation of H$_2$ and surface reducibility, as evidenced by the H$_2$-TPR of surface-oxidized catalysts (Supplementary Fig. 28). In addition, the H$_2$-TPD profiles of Pd/CNT and Pd-C/CNT are nearly identical (Supplementary Fig. 29), indicating the ligand shows limited influence to the site for H$_2$ adsorption/desorption. These results suggest that ligand on the catalyst does not block a specific site of deep hydrogenation as reported conventionally[53], and the site for H$_2$ activation lies likely on all the Pd domain. Therefore, the weakened interaction between Pd and reactants, other than site blocking, should play the major role in inhibited deep hydrogenation of C=C bond in ethylene.

The adsorption behaviors of DPE and ethene over clean Pd and Pd modified by alkenyl carbon (denoted as Pd-C) were also investigated by DFT modeling. The five-membered metallacycle was used as a representative alkenyl-type carbon ligand. Since the catalysts also show distinct adsorption behaviors in the absence of hydrogen (vide supra), the modeling does not involve dissociative H. Compared with clean Pd, the interaction distances and adsorption energies between adsorbed DPE or ethene and Pd-C (Fig. 4d, e) are longer and lower, respectively (Table 1). In addition, the C–C bond lengths in the aromatic ring and bond dihedral over Pd-C are comparable to that of free DPE (Table 1 and Supplementary Fig. 30). Similarly, the ethene on Pd-C also displays a lower degree of structural alteration (Table 1). It should be noted that the alkenyl ligand makes adsorption of DPE in the adjacent region highly unstable and the location of DPE is changed to a site with lower energy (Fig. 4d). By anchoring the alkenyl ligand at remote sites, the influence of an alkenyl ligand on the adsorption behavior of DPE at same adsorption location was also examined (Supplementary Fig. 31), which also shows less structural change of DPE over Pd-C than clean Pd. Besides the electronic modification to catalyst surface as indicated by XPS (Fig. 3c) and XAS (Supplementary Fig. 21), steric effect is also a possible mechanism to regulate the

selectivity[54]. Over 5Pd-C/Fe-C, benzene contributes up to 55.0% of the monomers in the products, and this number is 39.5% on 5Pd/Fe and 43.9% on 5Pd/Fe-C (Supplementary Fig. 6). Since 50% of benzene selectivity is expected in the primary products, the fact that 55.0% of benzene selectivity is observed over 5Pd-C/Fe-C indicates the preservation of majority of aromatic ring in benzene, as well as potential subsequent hydrodeoxygenation of phenol to benzene. In contrast, appreciable amounts of aromatic ring in benzene are hydrogenated over 5Pd/Fe and 5Pd/Fe-C. This result implies that carbon ligand inhibits the hydrogenation of not only large molecule (i.e., DPE), but also small one (i.e., benzene). Therefore, the steric effect may not play a major role in this selective catalysis. The experimental and theoretical results presented above suggest that the alkenyl ligand causes an electronic perturbation, possibly combined with steric effects, on the catalyst surface. This weakened the adsorption of both the aromatic ring and ethene[1]. The weakened adsorption of DPE and ethene favors its desorption from surface and thus prevents its activation for deep hydrogenation[55].

**Table 1 | Adsorption energies and changes in the structures of adsorbates onto different surfaces**

| Adsorbate/ surface | Adsorption energy (eV) | Structural changes (S$_{molecule, ads}$ – S$_{molecule, gas}$) | |
|---|---|---|---|
| | | Change in C–C bond length (Å) | Change in dihedral of C–H bond (°) |
| DPE/Pd | –1.10 | 0.043 | –21.7 |
| DPE/Pd-C | –0.86 | 0.017 | –10.2 |
| Ethene/Pd | –0.60 | 0.058 | –19.1 |
| Ethene/Pd-C | –0.50 | 0.053 | –17.9 |

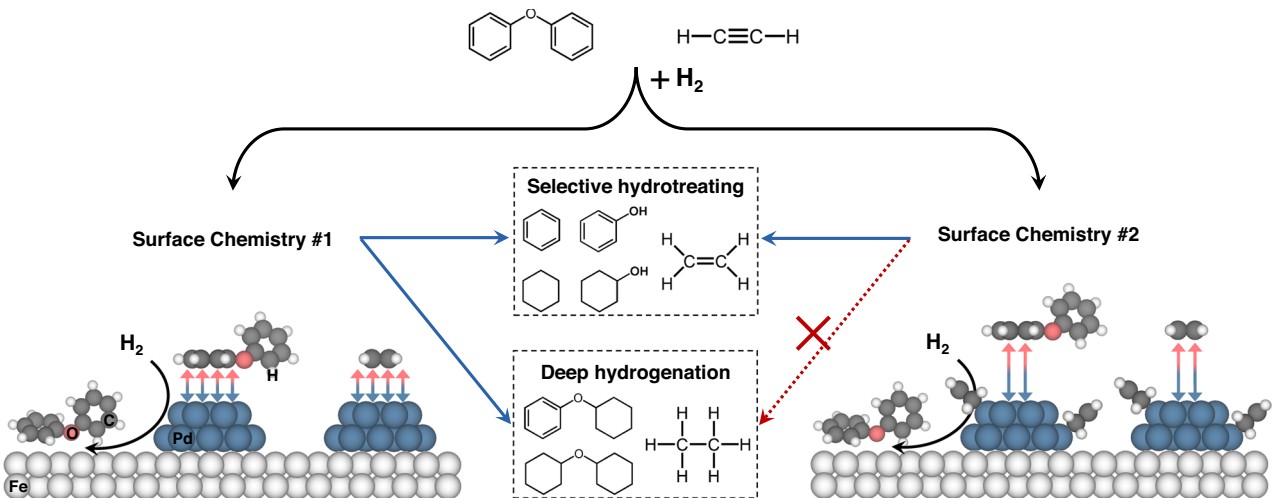

**Fig. 5 | Schematic overview of the working mechanisms for the selective hydrogenolysis of DPE and selective hydrogenation of acetylene over a catalyst coordinated with alkenyl carbon.** Pd facilitates the completion of C–O bond cleavage on Fe sites and tends to hydrogenate the sp²-hybridized carbon (Surface Chemistry #1). Coordinative alkenyl-type carbon species on Pd tailor the electronic structure of Pd and/or sterically inhibits the adsorption of DPE and ethene, leading to weakened interaction between Pd and sp²-hybridized carbon to inhibit deep hydrogenation (Surface Chemistry #2). Pd, Fe, C, O, and H atoms are depicted in blue, white/large, gray, red, and white/small, respectively.

## Discussion

Based on the above results, the working mechanism in selective C–O bond cleavage and selective semi-hydrogenation of acetylene over 5Pd-C/Fe-C is inferred as shown in Fig. 5. In hydrotreating of DPE, Fe domain selectively cleaves the C–O bond while the Pd domain activates H₂ which can be transferred to Fe to enhance the reductive elimination of the radicals from C–O bond cleavage. This bifunctionality stabilizes the Fe domain to remain in a highly active state. However, the characteristic property of bulk Pd also makes it intensely perturb and activate the sp²-hybridized carbon in aromatic ring, leading to direct hydrogenation of DPE. Similarly, in the hydrogenation of acetylene, bare Pd strongly interacts with sp²-hybridized carbon in the semi-hydrogenation product (i.e., ethene), facilitating secondary/ deep hydrogenation (Surface Chemistry #1 in Fig. 5). The homogeneous-like alkenyl ligand donates electrons to Pd to tailor the electronic property that elongates the distance and weakens interaction between the Pd and sp²-hybridized carbon in the reactants, leading to the inhibition of deep saturation (Surface Chemistry #2 in Fig. 5). Moreover, this unique type of ligand shows no inhibition to the rate of desired reaction pathway, that is distinct from the reported alkanethiolate-modified Pd on which a dramatic drop of production rate of desired products was observed[53]. These characters make alkenyl ligand a promising surface modifier for tunning the hydrogenation chemistry. It should be noted that, though the carbon effect has been recognized in several reactions such as Fischer-Tropsch process[56], various types of carbon (carbidic, graphitic, chain-like, etc.) and molecular structures co-exist in majority of the studies, making the proposed structure of active site largely speculative. In the present work, by controlling the concentration of carbon precursor gas (i.e., CO) in H₂ and performing a H₂ treatment, we have selectively removed the carbon spectators and identified the nature of active sites with a combination of molecular-level characterizations and DFT modeling.

In summary, two common properties of noble metals during hydrotreating reactions are activation of H₂ and hydrogenation of unsaturated bonds. Our study presents an efficient approach using homogeneous-like surface Pd-alkene metallacycle on a heterogeneous Pd catalyst to tailor the electronic interaction between the sp² carbon in reactants and surface Pd so that the deep hydrogenation is inhibited. Moreover, the coordinative alkenyl ligand shows limited influence on activation of H₂ on Pd and C–O bond cleavage on Fe. This enables the selective utilization of the Pd character with inhibited negative effect in the reaction. In the model reactions, i.e., hydrotreating DPE and hydrogenation of acetylene, above 90% selectivity to C–O bond cleavage and to semi-hydrogenation towards ethene were achieved with maintained productivity. This work sheds light on the controlled synthesis of selective hydrotreating catalysts via mimicking homogeneous analogs.

## Methods

### Synthesis of catalysts

The Fe₂O₃ was prepared with a precipitation method. Typically, (NH₄)₂CO₃ (Sigma-Aldrich, 1.5M) solution was added dropwise to a solution of Fe(NO₃)₃ (Sigma-Aldrich, 3M) under stirring, forming a dark crimson slurry. The precipitate was washed with Milli-Q water via filtration until the pH reached 8, and then transferred into a convection oven for drying at 80 °C overnight. The obtained solid sample was then crushed and sieved to <100 mesh, followed by a calcination at 400 °C for 5 h to obtain the Fe₂O₃. The Fe₂O₃ was then impregnated with Pd using aqueous solution of Pd(NH₃)₄(NO₃)₂ (Sigma-Aldrich, 10 wt% solution with 99.99% metal based purity) of varied concentrations. The diluted solution of calculated concentrations was then added dropwise to Fe₂O₃ powder, followed by drying at 80 °C overnight in air. The weight percentages of Pd in Pd/Fe₂O₃ samples were 0.1 wt%, 1 wt%, and 5 wt% of Pd.

The above oxide precursors were reduced with UHP H₂ (50 mL/ min) at 350 °C (2 h for 1 wt% and 5 wt% samples, 3 h for 0.1 wt% sample and bare Fe₂O₃). After cooling to ambient temperature in flowing UHP N₂ (50 mL/min), the sample was passivated by 1 vol% O₂/N₂ (50 mL/ min) for 2 h. The obtained samples are denoted as Fe, 0.1 Pd/Fe, 1 Pd/ Fe, and 5 Pd/Fe, respectively. 5Pd-C/Fe-C was prepared with same procedure as 5 Pd/Fe except using 0.4 vol% CO/H₂ (50 mL/min) instead of UHP H₂. The sample was then treated with 10 vol% H₂/He (50 mL/ min) at 250 °C for 0.5 h, followed by a passivation with 1 vol% O₂/N₂ (50 mL/min) for 2 h. To synthesize 5 Pd/Fe-C, the oxide precursor was first reduced in 0.4 vol% CO/H₂ (50 mL/min) for 1.5 h, then UHP H₂ (50 mL/min) for 1 h to selectively remove the carbon on Pd.

The impregnation process to synthesize bare Pd/CNT (i.e., 5 wt% Pd supported on carbon nanotube) was the same as that of 5 Pd/Fe except using carbon nanotube as support. The dried sample was then reduced with UHP H₂ (50 mL/min) at 350 °C for 2 h. Pd-¹³C/CNT

(i.e., Pd/CNT decorated by $^{13}C$-labeled carbon) was prepared with same procedure as Pd/CNT except using 0.4 vol% $^{13}CO$ (99% abundancy, Cambridge Isotope Laboratories, Inc.) balanced by H$_2$ instead of UHP H$_2$ in the reduction process. The sample was then pretreated with 10 vol% H$_2$/He (50 mL/min) at 250 °C for 0.5 h. Both catalysts were passivated using the same procedure as described above.

## Characterizations

HADDF-STEM with EDS elemental mapping was conducted on FEI Tecnai G2 F20 S-TWIN STEM operated at 200 kV. In the preparation of the sample, a dry powder was dispersed on a lacey-carbon coated 200 mesh Cu grids.

H$_2$-TPRea was carried out on Chemisorption Analyzer (Micromeritics AutoChem 2920) equipped with a quadrupole mass spectrometer (Omnistar gas analyzer GSD 301). The in situ synthesized sample was pretreated with UHP H$_2$ (50 mL/min) at 250 °C for 1 h to simulate the surface during the hydrotreating reaction. Then the temperature was ramped to ambient followed by switching gas to 10 vol% H$_2$/Ar. After purging for 20 min, the temperature was ramped to 700 °C (10 °C/min). C$_2$H$_4$-TPD was performed on Chemisorption Analyzer (Micromeritics AutoChem 2920) equipped with a mass spectrometer (Omnistar gas analyzer GSD 301). In each test, the catalyst was first pretreated with 10 vol% H$_2$/Ar (40 mL/min) at 150 °C, followed by cooling to −30 °C in UHP Ar (40 mL/min). After soaking in 10 vol% C$_2$H$_4$/He for 30 min and purged for 60 min, the temperature was ramped to 300 °C.

NEXAFS spectroscopy was performed in the ultrahigh vacuum end station attached to the horizontally polarized soft X-ray spectroscopy beamline in Australian synchrotron. The C K-edge spectra were collected at angle of 55° between the direction vector of the beam and surface plane of sample, which avoided the polarization effect of synchrotron radiation. The surface-sensitive TEY mode was applied in the measurement.

Pseudo-in situ XPS measurements were carried out after pseudo in situ processing under 100 mL/min UHP H$_2$ at 250 °C for 0.5 h. XPS measurements were performed with a Physical Electronics Quantera Scanning X-ray Microprobe. Charge referencing was made using the binding energy for Fe $2p_{3/2}$ at 706.8 eV[12].

$^{13}C$ MAS solid-state NMR experiments were performed on a Varian-Agilent Inova widebore 300 MHz NMR spectrometer with a 4 mm rotor. $^1H$–$^{13}C$ cross-polarization NMR was used to monitor surface carbon species by enhancing signals by transferring magnetization of $^1H$ to $^{13}C$. The 4 mm rotor with about 20 mg of loaded catalysts was spined at a rate of 7 kHz with a contact time of 1 ms, a recycle delay of 2 s, and a pulse width of 3.25 μs. 25,000 scans were collected for each spectrum.

In situ ATR-FTIR was carried out to study the adsorbed DPE over the surface of catalysts. The experiments were conducted on Bruker Tensor II spectrometer and a custom-designed ATR cell. The as-prepared catalyst was ground into fine powder so that it could be suspended in hexane by sonication. Then the catalyst was deposited on the ZnSe Internal Reflection Element by dip coating. The thin layer of catalyst coating was then pretreated in 10%H$_2$/He (40 mL/min) at 250 °C for 1 h to simulate the surface during reaction, followed by cooling down to 100 °C. After N$_2$ purging for 1 h, a background was collected. Then DPE vapor (10 Pa, balanced by N$_2$, 40 mL/min) was introduced to the catalyst. The spectra were collected by averaging 64 scans at a resolution of 4 cm$^{-1}$. For the spectra obtained under H$_2$ cofeeding, the background was collected in 10%H$_2$/He atmosphere and the DPE vapor was balanced with 10%H$_2$/He (40 mL/min).

## Evaluation of catalytic performance

The hydrotreating reaction was performed in a stainless-steel Parr reactor (Series 4560, 300 mL). In a typical measurement, 0.03 g of catalyst, 1.08 g of DPE (1 mL, Sigma-Aldrich, 99%), and 50 mL of hexadecane (Sigma-Aldrich, 99%) were loaded into a glass liner, which was transferred to the stainless-steel reactor. After purging with 4 MPa

H$_2$ for 3 times, the temperature was ramped to 250 °C (15 °C/min) followed by pressurizing with H$_2$ to 5.8 MPa where the stirring (800 rpm) of the mixed slurry solution and reaction started. After the reaction was completed, the products were analyzed by a gas chromatograph (GC, Agilent 7890A) equipped with a DB-FFAP column (30 m, 0.32 mm, 0.25 μm) and flame ionization detector (FID). In the stability test, the catalyst was directly used for the next cycle of reaction. No further treatment was conducted after removing the liquid mixture. The TOFs (measured with conversions below 20%) and selectivity were defined as follows: TOF of direct C−O bond cleavage = (moles of produced monomers/2)/moles of exposed Fe/reacting time; TOF of direct hydrogenation = moles of produced ring-saturated dimer/moles of exposed Pd/reacting time; selectivity [%] = (moles of carbon in the specific product/moles of carbon in all products) × 100%. The carbon balance was in 95–105%.

Hydrogenation of acetylene was conducted with a continuous flow reactor at 150 °C and ambient pressure. Prior to the activity test, 0.03 g of catalyst that was mixed with 0.3 g of SiC was pretreated in 8 vol% H$_2$/N$_2$ (60 mL/min) at 150 °C for 1 h. Thereafter, 4% C$_2$H$_2$−8% H$_2$/N$_2$ (60 mL/min) was introduced to the catalyst bed to start the hydrogenation reaction. A customized gas chromatograph (Shimadzu GC-2014) with HP-Plot Q column (30 m, 0.53 mm, 40 μm), flame ionization detector (FID), and thermal conductivity detector (TCD) were used to analyze the products. Acetylene conversion and product selectivity are calculated as follows: conversion = (moles of acetylene in the inlet − moles of acetylene in the outlet)/moles of acetylene in the inlet; selectivity [%] = (moles of carbon in the specific product/moles of carbon in all products) × 100%.

## Computational methods

All computational studies were based on density functional theory (DFT) and were performed with the Gaussian 16 series of programs package[57]. Since the B3LYP/Lanl2DZ computational level has been proved to be reliable for systems containing organic molecules and Pd[58,59], it was used throughout the calculations. The NMR chemical shifts were computed by applying the GIAO model. The details of methods and models are described in the Supplementary Information.

## Data availability

The data supporting the findings of this study are available within the paper and its Supplementary Information files. Should any raw data files be needed in another format they are available from the corresponding authors upon reasonable request. Source data are provided with this paper.

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

## Acknowledgements

This work was supported by the U.S. Department of Energy (DOE), Office of Science, Basic Energy Sciences (BES) and Division of Chemical Sciences, Geosciences, and Biosciences within the Catalysis Science program (DE-AC05-RL01830, FWP-47319). NMR experiments were performed in the Environmental Molecular Sciences Laboratory (EMSL) (grid.436923.9), a DOE Office of Science User Facility sponsored by the Office of Biological and Environmental Research and located at Pacific Northwest National Laboratory (PNNL). PNNL is a multiprogram national laboratory operated by Battelle for the U.S. Department of Energy under Contract DE-AC05-76RL01830. J.Z., C.Z., and H.H. thank the financial supports from National Natural Science Foundation of China (22025604) and Science and Technology Innovation and Development Center, CAS (HT2022100315). Monash Centre for Electron Microscopy, beamline BL14W1 of Shanghai Synchrotron Radiation Facility (SSRF), soft X-ray beamline in Australian Synchrotron and beamline BL16A1 of Taiwan National Synchrotron Radiation Research Center (NSRRC) are also acknowledged.

## Author contributions

J.Z. and J.S. conceived the research ideas and designed the experiments. Y.W., C.Z., and J.S. supervised and led the project. J.Z. synthesized and evaluated the catalysts, performed the HRTEM, temperature-programmed experiments, ATR-FTIR characterizations and DFT modeling. W.H. and J.H. carried out the NMR characterizations. B.Q. and L.Z. conducted the STEM, XAS, and NEXAFS characterizations. M.H.E. performed the XPS experiments. H.L. and B.S. carried out the acetylene hydrogenation reaction and TEM measurement. J.Z., W.H., J.S., C.Z., H.H., and Y.W. wrote the manuscript. All authors discussed the results, contributed towards data interpretation and commented on the manuscript.

## Competing interests

The authors declare no competing interests.
