## [Peer Review File · Nature Communications]

Tuning Hydrogenation Chemistry of Pd-based Heterogeneous Catalysts by Introducing Homogeneous-like LigandsREVIEWER COMMENTS

Reviewer #1 (Remarks to the Author):

This manuscript reports that the selective reaction of DPE to the C–O cleaved species and of C₂H₂ to C₂H₄ were achieved by deposition of carbon on Pd-Fe catalysts, and that the state of deposited carbon was thoroughly analyzed and concluded to be Pd-alkene metallacycle state. The selectivity improvement without degradation of activity is important and interesting achievement. The idea to add the homogeneous-catalyst-like metallacycle ligand is highly original. The CO/H₂ treatment for adding the ligand is very simple way; however, it is not easy to find an appropriate treatment condition to make a desired carbon as the carbon state was sensitive to CO concentration. This fact also indicates a uniqueness of this study. The authors' claim about mechanisms is strongly supported by many experiments including ones using various spectroscopic methods and by DFT calculations. Since I'm not familiar with details of DFT calculation, if validity and reliability are verified by other reviewers who are expert in DFT calculation, I believe this manuscript is worthy for publication in Nature Communications after minor revision that should consider following comments.

Comments (basically arranged in the order of pages)

1. "hydrodeoxygenation" in the caption of Fig. 1b is probably wrong.
2. Readers can imagine the state of samples more easily if there is a Table including surface area and active site/area of each sample.
3. What is the XRD peak around 36 deg in Fig. S4? (also two peaks around 60 deg for 1Pd/Fe)
4. How to use "diffraction indices" and "Miller indices" is wrong. For example, "200 peak" means "the second order reflection of 100 reflection". From (100) plane, 100, 200, 300, 400,..., n00 peaks can be observed. In XRD pattern, "200 Pd" without parentheses is correct description. Also, the authors wrote "[111] lattice plane" and "[110] plane". "[hkl]" describes "direction" and "(hkl)" describes "plane". Description in Fig. S14 and S15 also must be corrected.
See paragraph below eq. (13) in <https://link.springer.com/article/10.1007/s42452-020-2498-5>
5. I checked Ref. 23 but cannot understand why shoulder at 2.2 Å is contributed by Pd-Fe scattering. More explanation would be helpful.
6. In page 9, "the C species in 5Pd/Fe-C is mainly on Fe region"
I think the C species are uniformly distributed on both Pd and Fe regions in Fig. 2, though I agree that the amount of C on Pd in 5Pd/Fe-C is much less than that in Pd-C/Fe-C as bright spots of C can be seen on Pd in Pd-C/Fe-C in Fig. 2d.
7. In Supplementary Note 5, Fig. S18 → Fig. S17, Fig. S23b → Fig. S22b.

8. 361 C in Fig. 3a corresponds to 332 C in Fig. S17. What's the origin of difference as much as 30 C?

9. In p. 11, "Using 5Pd/Fe as background (Supplementary Fig. S19)".

Fig. S19 → Fig. S18.

10. In Supp Note 8, Fig. S25a → Fig. S24a, Fig. S10c → Fig. S24c.

11. In page 15, "It suggests the role of ligand on the catalyst is not blocking specific site of deep hydrogenation as reported conventionally"

I think it's better to add a reference for "reported conventionally". Also, I can understand no site blocking for H₂ from experimental results, but for C₂H₄, there is a possibility that C blocks a site for strong adsorption of C₂H₄ and leads to C₂H₄ adsorption on weak adsorption site. If you say "no site blocking for C₂H₄", Fig. 4e may be helpful since C₂H₄ adsorbs on the same site with the same configuration in two catalysts.

12. Table 1 uses "binding energy" with minus value, but text uses "adsorption energy". Maybe, "adsorption energy" is better in Table 1.

13. In the caption of Fig. S28, "one Pd particle" may be "on Pd particle".

14. In page 16, "These results suggest an electronic perturbation of the catalyst surface by the alkenyl ligand and thus weakened adsorption of the aromatic ring and ethene"

Besides electronic perturbation, how about interaction between adsorbed molecules and ligands since ligands are crowded in Fig. S28. Also, how about steric effect (or site blocking) for DPE by crowded ligands? Actually, in Fig. S6, 5Pd-C/Fe-C completely hydrogenated phenol and hydrogenated 30% of benzene to cyclohexane, which indicates that larger molecules cannot adsorb but smaller ones can adsorb.

15. The bottom left in Fig. 5, the C–O broken products are also obtained; thus, the current description is somewhat inappropriate. I understand what the authors want to say, but it is better if the description is improved.

16. I believe this study is highly valuable. However, I feel the title "by mimicking homogeneous analogues" is inappropriate, because the active site is not Pd inside of metallacycle but far from the ligands. Thus, I recommend to change the title; for example, "Tuning hydrogenation chemistry of Pd-based heterogeneous catalysts by introducing homogeneous analogue ligands".

Reviewer #2 (Remarks to the Author):

The paper includes a very extensive work with the use of numerous techniques, however, the results do not clearly evidence the higher electron-density of Pd in 5Pd-C/Fe-C and its origin from carbonaceous

compounds. Moreover, the benefit of a high electron-density of Pd for hydrogenation reactions in general cannot be claimed as new because it is already reported for other hydrogenation reactions. The following are the main points which need revision:

- Firstly, the use of alkenyl-type ligands to induce high electron-density to Pd for C-O bond cleavage do not seem to be justified because though selectivity is greatly increased, conversion is drastically reduced and therefore the yield to C-O bond cleavage products do not improve (Figure S.5).
- If Pd has a higher electron density in 5Pd-C/Fe-C catalysts, a displacement of normalized absorption with energy should be observed by Pd K edge XANES with respect the foil and the other catalysts. This is not clearly observed in Figure S.12. To better find out if this displacement exists would be necessary to represent XANES first derivative as a function of energy.
- Pd dispersion values and catalysts surface area are not found in the manuscript. They should be reported. In view of XRD results (Fig. S14), it seems that 5Pd-C/Fe-C shows the smallest Pd particles, it is reported in the bibliography that smallest Pd particles are prone to present a higher electron-density. To rule out the effect of metal particle sizes in the electronic properties of Pd, correlation of all these properties and catalytic ones must be analyzed in detail.
- Carry on with the effect of Pd particle size, is found in the literature that small Pd particle sizes favour selectivity to ethylene in other hydrogenation reactions, therefore the influence of Pd particle size in the target reactions of the manuscript should be studied in detail. In that sense, as stated above, metal particle sizes should be clearly reported for all the catalysts, and it would be worthy to determine metal particle sizes from TEM and/or DRX measurements.
- It is reported in the manuscript that in STEM images carbon compounds are not associated to Pd in Pd/Fe-C (Fig 2.c), but reviewer do not agree with this assertion because although with less intensity, a higher intensity of C patterns in the areas of brightness of Pd is observed.
- Is there any alloy Pd-Fe formation? It is know that the addition of a second metal, and particularly Fe, can modify electronic properties of Pd. Moreover, the addition of a carbon treatment could modify the reduction atmosphere and modify the interaction of both metals modifying the effect of Fe on Pd particles instead being the carbonaceous species properly those modifying electronic properties of Pd. ¿Why can be this effect discarded?
- To better rule out the effect of Fe in the electronic properties of Pd a Pd-Fe/CNT catalysts should be also studied in addition to Pd/CNT and Pd-C/CNT.
- ¿Why does products distribution change in such a significant extent for 5Pd-C/Fe-C?
- As NMR has been performed without Fe in the catalysts, the findings are questionable as the catalyst change.
- The position of figure 2 within the text is not suitable.
- The description of preparation of CNT supported Catalysts (lines 392-396 of page 20) is not clear. It is confusing which catalysts have been prepared.
- A comparison of activity and selectivity of Pd-C/CNT and 5Pd-C/Fe-C should be included.

Reviewer #1 (Remarks to the Author):

This manuscript reports that the selective reaction of DPE to the C–O cleaved species and of C₂H₂ to C₂H₄ were achieved by deposition of carbon on Pd-Fe catalysts, and that the state of deposited carbon was thoroughly analyzed and concluded to be Pd-alkene metallacycle state. The selectivity improvement without degradation of activity is important and interesting achievement. The idea to add the homogeneous-catalyst-like metallacycle ligand is highly original. The CO/H₂ treatment for adding the ligand is very simple way; however, it is not easy to find an appropriate treatment condition to make a desired carbon as the carbon state was sensitive to CO concentration. This fact also indicates a uniqueness of this study. The authors' claim about mechanisms is strongly supported by many experiments including ones using various spectroscopic methods and by DFT calculations. Since I'm not familiar with details of DFT calculation, if validity and reliability are verified by other reviewers who are expert in DFT calculation, I believe this manuscript is worthy for publication in Nature Communications after minor revision that should consider following comments.

Comment 1: “hydrodeoxygenation” in the caption of Fig. 1b is probably wrong.

Response: We thank the reviewer for the positive comments and helpful suggestions. We also appreciate for pointing out this confusing statement. “hydrodeoxygenation” in the caption of Fig. 1b should be “hydrogenation”.

Actions taken: We revised the caption of Fig. 1b.

“TOFs of C-O bond cleavage (a, based on exposed Fe) and direct hydrogenation (b, based on exposed Pd) over different catalysts (at < 20% conversion).”

Comment 2: Readers can imagine the state of samples more easily if there is a Table including surface area and active site/area of each sample.

Response and actions taken: We have added a table summarizing the surface area, amounts of exposed Pd and Fe, as well as the dispersion data in the Supplementary Information. It is discussed on page 6 in the main manuscript.

Supplementary Table S1 Structural properties of the catalysts.

	Surface area (m ² /g)	Number of exposed Pd (mmol/g) ^a	Number of exposed Fe (mmol/g)	Dispersion of Pd (%) ^a	Particle size of Pd (nm) ^a	Particle size of Pd (nm) ^b
Fe	8.2	\	0.19	\	\	\
0.1Pd/Fe	8.1	0.011	0.18	81.7	1.4	-
1Pd/Fe	8.0	0.027	0.15	20.2	5.5	-
5Pd/Fe	9.2	0.055	0.16	8.8	12.7	10.8
5Pd/Fe-C	9.2	0.052	0.16	8.3	13.4	11.2
5Pd-C/Fe-C	9.4	0.048	0.17	7.7	14.5	10.7

a) Determined by CO pulse chemisorption.

b) Determined by TEM.

“..... the calculated turnover frequency (TOF) of C-O bond cleavage and direct hydrogenation were based on exposed Fe and Pd, respectively (details are shown in Supplementary Method, Fig. S3 and Table S1).”

Comment 3: What is the XRD peak around 36 deg in Fig. S4? (also two peaks around 60 deg for 1Pd/Fe)

Response: Thank you for pointing these out. The peaks at 35.4, 56.9 and 62.5 are characteristics of Fe_3O_4 that may lie in the core of catalyst particle. It should be noted that the all the catalysts were reduced under same conditions except that 5Pd/Fe and 1Pd/Fe were reduced with less time than Fe and 0.1Pd/Fe (Page 21 of the main manuscript). Therefore, the core of 1Pd/Fe may not be fully reduced. We performed the additional in-situ Raman spectroscopy of these samples pretreated in H_2 at 250 °C for 0.5 h. Compared with the oxide reference (i.e., the untreated Fe with oxide overlayer), no oxide peak was observed in the spectra of Fe, 0.1Pd/Fe and 1Pd/Fe. It indicates the Fe species in surface layer should be still in metallic state in the hydrotreating reaction. This is consistent with the in-situ XPS results shown in our previous study (ACS Catal. 2014, 4, 3335–3345).

Actions taken: The Raman spectra and annotation have been added in Supplementary Fig. S4.

Supplementary Fig. S4 XRD patterns (a) and Raman spectra (b) of catalysts containing different amounts of Pd. The Raman spectra of Fe, 0.1Pd/Fe and 1Pd/Fe were taken after an in-situ pretreatment using 10% H_2/N_2 for 0.5 h.

The small peaks at 35.4, 56.9 and 62.5° are attributed to the diffraction of Fe_3O_4 that may be located inside the core of catalyst particle. It should be noted that the all the catalysts were reduced under same conditions except that Fe and 0.1Pd/Fe were reduced at longer time. Therefore, the core of 1Pd/Fe may not be fully reduced. In addition, compared with the oxide reference (i.e., the untreated Fe with oxide overlayer), no oxide peak was observed in the Raman spectra of Fe, 0.1Pd/Fe and 1Pd/Fe. This indicates that the Fe species on surface should still be in metallic state in the hydrotreating reaction, which is consistent with the in-situ XPS results in our previous

study¹³.

Comment 4: How to use “diffraction indices” and “Miller indices” is wrong. For example, “200 peak” means “the second order reflection of 100 reflection”. From (100) plane, 100, 200, 300, 400, ..., $n00$ peaks can be observed. In XRD pattern, “200 Pd” without parentheses is correct description. Also, the authors wrote “[111] lattice plane” and “[110] plane”. “[hkl]” describes “direction” and “(hkl)” describes “plane”. Description in Fig. S14 and S15 also must be corrected.

See paragraph below eq. (13) in <https://link.springer.com/article/10.1007/s42452-020-2498-5>

Response: We thank the reviewer for pointing out the misleading terminologies. We have carefully referred the article and corrected those in Fig. S4, S14 and S15 (S4, S15 and S16 in the revised version), as well as the corresponding discussions.

Actions taken: We have corrected the supplementary figures and the corresponding discussions in both main manuscript and Supporting Information.

On page 8 in the main manuscript, “The measured interplanar spacing for the lattice fringes of the supported particles is ~0.22 nm which corresponds to the (111) plane of face-centered cubic Pd²⁵. The support has an interplanar distance of ~0.21 nm, typically attributed to (110) plane of body-centered cubic Fe²⁶.”

On page 19 in Supporting Information, “All the catalysts display patterns containing peaks at 44.7 ° and 65.0 °, that can be indexed to 110 and 200 diffractions of metallic Fe with body-centered cubic crystal structure (JCPDS no. 06-0696). The main diffraction peak corresponds to (110) facet, that is aligned with the HRTEM showing the support has a lattice fringe of 0.21 nm (Fig. 2a).²³ No diffractions of carbide of Fe^{24,25} can be detected. Besides the peaks attributed to Fe, others with low intensity also appear at 40.1 ° and 46.7 °, which are the 111 and 200 diffractions of metallic Pd with face-centered cubic crystal structure (JCPDS no. 46-1043).”

Supplementary Fig. S4 XRD patterns (a) and Raman spectra (b) of catalysts containing different amounts of Pd. The Raman spectra of Fe, 0.1Pd/Fe and 1Pd/Fe were taken after an in-situ

pretreatment using 10% H₂/N₂ for 0.5 h.

Supplementary Fig. S15 XRD patterns of as-synthesized 5Pd/Fe, 5Pd/Fe-C and 5Pd-C/Fe-C.

Supplementary Fig. S16 XRD patterns of spent catalysts collected after hydrotreating reaction.

Comment 5: I checked Ref. 23 but cannot understand why shoulder at 2.2 Å is contributed by Pd-Fe scattering. More explanation would be helpful.

Response and Actions taken: The previous discussion of the shoulder peak at 2.2 Å is not appropriate. We revised it and updated with the references (J. Catal. 2013, 306, 47-57; Catal. Today 2019, 323, 123-128).

“Compared with Pd foil, spectra of all catalysts display a decay of the main peak, leading to the emergence of the abreast peak at ~2.2 Å. The similar change has been widely reported for the Pd-Fe bimetallic system and ascribed to the interference between Pd-Pd and Pd-Fe oscillations^{23,24}.”

Comment 6: In page 9, “the C species in 5Pd/Fe-C is mainly on Fe region”

I think the C species are uniformly distributed on both Pd and Fe regions in Fig. 2, though I agree that the amount of C on Pd in 5Pd/Fe-C is much less than that in Pd-C/Fe-C as bright spots of C can be seen on Pd in Pd-C/Fe-C in Fig. 2d.

Response: We agree with the reviewer’s comment. Indeed, we mentioned that “carbon is detected on both Pd and Fe domains of 5Pd/Fe-C” in the discussion of STEM images. The sentence referred by the reviewer was trying to elucidate the structural difference of the three catalysts. To avoid the confusion, we rearranged the layout of Fig. 2 and rephrased the corresponding discussion.

Actions taken: We changed the layout of the previous Fig. 2b-d to current Fig. 2b-e. The corresponding discussion on Page 9 was revised.

Fig. 2. Microscopic analysis of Pd/Fe, Pd/Fe-C and Pd-C/Fe-C. a, HRTEM images of samples, the left and right ones show the regions in white squares; b-d, STEM images and EDS elemental mapping of Pd, C, Fe over 5Pd/Fe (b), 5Pd/Fe-C (c) and 5Pd-C/Fe-C (d), Brown: Pd, Green: C, Navy: Fe; e, schematic illustration of carbon locations on Pd and Fe. All samples were pretreated with 10%H₂/Ar at 250 °C for 1 h to simulate the catalysts after the hydrotreating reaction.

“Based on the above results, structures of the three catalysts are proposed and schematically presented in Fig. 2e. Compared with the clean 5Pd/Fe and the 5Pd/Fe-C that show C on both Pd and Fe region, the Pd in 5Pd-C/Fe-C is covered with a significantly higher amount of C species.”

Comment 7: In Supplementary Note 5, Fig. S18 → Fig. S17, Fig. S23b → Fig. S22b.

Response and actions taken: We thank the reviewer for pointing out these mistakes. Supplementary Note 5 has been revised accordingly. The figure numbers have been updated in the revised version.

“In Supplementary Fig. S18, the sample with carbon ligand obtained by the same synthesis procedure as 5Pd-C/Fe-C shows the methane formation peak at 332 °C,”

“As shown in Supplementary Fig. S25b, Pd microparticle shows negligible carbon deposition,”

Comment 8: 361 C in Fig. 3a corresponds to 332 C in Fig. S17. What’s the origin of difference as

much as 30 C?

Response: The support of catalyst shown in Fig. 3a is Fe, while the one in Fig. S17 (Fig. S18 in the revised version) is CNT. The different interactions between Pd and supports (i.e., Fe and CNT) would lead to different modifications on Pd and thus influence the temperature needed to hydrogenate the carbon species on it.

Comment 9: *In p. 11, “Using 5Pd/Fe as background (Supplementary Fig. S19)”. Fig. S19 → Fig. S18. In Supp Note 8, Fig. S25a → Fig. S24a, Fig. S10c → Fig. S24c.*

Response and actions taken: We have corrected these mistakes.

“Using Fe for charge correction (Supplementary Fig. S20) and 5Pd/Fe as background, C 1s XPS difference spectrum of 5Pd-C/Fe-C displays a peak located at 284.1 eV that has been attributed to a chain hydrocarbon bonded to metal (Supplementary S19 and Note S6)”

“In Supplementary Fig. S27a, three peaks appear at 1582, 1483 and 1456 cm^{-1} , Magnification of ν_8 band shows that it presents an asymmetric shape, as shown in Fig. S27b. We therefore deconvolute this peak as shown in Fig. S27c.”

Comment 10: *In page 15, “It suggests the role of ligand on the catalyst is not blocking specific site of deep hydrogenation as reported conventionally”*

I think it's better to add a reference for “reported conventionally”. Also, I can understand no site blocking for H₂ from experimental results, but for C₂H₄, there is a possibility that C blocks a site for strong adsorption of C₂H₄ and leads to C₂H₄ adsorption on weak adsorption site. If you say “no site blocking for C₂H₄”, Fig. 4e may be helpful since C₂H₄ adsorbs on the same site with the same configuration in two catalysts.

Response: We have added the reference for “reported conventionally”. As for the possible site blockage by C and inhibition of strong adsorption of C₂H₄, we did think about this scenario. Assuming the strongly adsorbed C₂H₄ is inhibited by a specific site blockage, the C₂H₄-TPD would show the evidence, i.e., no desorption of strongly adsorbed C₂H₄ should be observed so that the shape of desorption peaks should be changed. However, all the C₂H₄ desorption profiles of the catalysts shown in Fig. 4c contain peaks having similar shape (i.e., two abreast peaks overlap with each other). It suggests the inhibition of deep hydrogenation should not be ascribed to site blocking. Instead, the lower temperature of desorption on 5Pd-C/Fe-C, compared with the other two catalysts, suggests that it should be owing to the weakened interaction of all kinds of adsorption.

Actions taken: We added the reference as mentioned, and the detailed analysis about C₂H₄-TPD on Page 16.

“It should be noted that the desorption peaks of 5Pd-C/Fe-C have comparable shape with the other two catalysts. It indicates that all the catalysts have similar sites for the adsorption of ethene, other than blocking of a specific kind of site by the coordinative carbon on 5Pd-C/Fe-C for the selective hydrogenation.”

“These results suggest that ligand on the catalyst does not block a specific site of deep hydrogenation as reported conventionally⁵³,.....”

Comment 11: Table 1 uses “binding energy” with minus value, but text uses “adsorption energy”. Maybe, “adsorption energy” is better in Table 1.

Response and actions taken: We agree with the reviewer to use consistent terminology “adsorption energy” to minimize confusions. This term in both table and caption has been revised.

Table 1. Adsorption energies and changes in the structures of adsorbates onto different surfaces.

Adsorbate/Surface	Adsorption energy (eV)	Structural changes ($S_{\text{molecule, ads}} - S_{\text{molecule, gas}}$)	
		Change in C-C bond length (Å)	Change in dihedral of C-H bond (°)
DPE/Pd	-1.10	0.043	-21.7
DPE/Pd-C	-0.86	0.017	-10.2
Ethene/Pd	-0.60	0.058	-19.1
Ethene/Pd-C	-0.50	0.053	-17.9

Comment 12: In the caption of Fig. S28, “one Pd particle” may be “on Pd particle”.

Response and action taken: We thank the reviewer for pointing out the typo. This caption has been revised accordingly.

“Supplementary Fig. S31 The influence on the interaction between Pd and aromatic ring by varying the concentration of metallacycle on Pd particle.”

Comment 13: In page 16, “These results suggest an electronic perturbation of the catalyst surface by the alkenyl ligand and thus weakened adsorption of the aromatic ring and ethene”

Besides electronic perturbation, how about interaction between adsorbed molecules and ligands since ligands are crowded in Fig. S28. Also, how about steric effect (or site blocking) for DPE by crowded ligands? Actually, in Fig. S6, 5Pd-C/Fe-C completely hydrogenated phenol and hydrogenated 30% of benzene to cyclohexane, which indicates that larger molecules cannot adsorb but smaller ones can adsorb.

Response: Thank you for the great question. The direct interaction between the DPE and ligands is also possible, which may affect the adsorption configuration of DPE. As discussed in our response to comment 10, the ligand should not block the active sites for adsorption. Indeed, we have discussed “Coordinative alkenyl-type carbon species on Pd tailor the electronic structure of Pd and/or sterically inhibits the adsorption of DPE and ethene, leading to weakened interaction between Pd and sp^2 -hybridized carbon to inhibit deep hydrogenation (Surface Chemistry #2)” in the caption of Fig. 5 on Page 18.

As for the secondary phenol hydrogenation, our previous study (Catal. Today 2020, 339, 305-311 and Chem. Sci. 2020, 11, 5874-5880) has demonstrated a different ring saturation mechanism. Namely, tautomerization of phenol opens another way to rapidly hydrogenate phenol even on pure Fe surface.

In fact, all the catalysts in this study completely hydrogenated/hydrodeoxygenated phenol to cyclohexanol and cyclohexane etc. (regardless of the presence of Pd, Fig. S6). Therefore, phenol hydrogenation cannot be used to evaluate the activity of direct ring hydrogenation on Pd domain.

More importantly, since the stoichiometry of C-O cleavage reaction is 1mol of DPE + 1mol of H₂ = 1mol of benzene + 1mol of phenol, selectivity to each product should be equal at 50% (formula to calculate selectivity is shown in Method section). However, in the products of 5Pd-C/Fe-C, benzene takes 55.0% of the monomers (i.e., 24.6% cyclohexane, 1.7% cyclohexene, 15.7% cyclohexanol and 51.3% benzene in Fig. S6, the “30% cyclohexane” mentioned by the reviewer is actually led by phenol hydrogenation and deoxygenation other than benzene hydrogenation). In contrast, it's 39.5% for 5Pd/Fe and 43.9% for 5Pd/Fe-C (see Fig. S6). It indicates that the aromatic ring in benzene is preserved, and a few phenol molecules are converted to benzene over 5Pd-C/Fe-C, while over 5Pd/Fe and 5Pd/Fe-C, appreciable amounts of aromatic ring in benzene are hydrogenated. This implies that the inhibition of hydrogenation by carbon ligand happens not only on large DPE, but also on small benzene. Therefore, it is inferred that, other than steric effect, electronic effect should play a major role in this selective catalysis.

Actions taken: We have added the analysis of the role of steric effect on page 17.

“Besides the electronic modification to catalyst surface as indicated by XPS (Fig. 3C) and XAS (Supplementary Fig. S21), steric effect is also a possible mechanism to regulate the selectivity⁵³. Over 5Pd-C/Fe-C, benzene contributes up to 55.0% of the monomers in the products, and this number is 39.5% on 5Pd/Fe and 43.9% on 5Pd/Fe-C (Supplementary Fig. S6). Since 50% of benzene selectivity is expected in the primary products, the fact that 55.0% of benzene selectivity is observed over 5Pd-C/Fe-C indicates the preservation of majority of aromatic ring in benzene, as well as potential subsequent hydrodeoxygenation of phenol to benzene. In contrast, appreciable amounts of aromatic ring in benzene are hydrogenated over 5Pd/Fe and 5Pd/Fe-C. This result implies that carbon ligand inhibits the hydrogenation of not only large molecule (i.e., DPE), but also small one (i.e., benzene). Therefore, the steric effect may not play a major role in this selective catalysis. The experimental and theoretical results presented above suggest that the alkenyl ligand causes an electronic perturbation, possibly combined with steric effects, on the catalyst surface. This weakened the adsorption of both the aromatic ring and ethene¹”.

Comment 14: *The bottom left in Fig. 5, the C–O broken products are also obtained; thus, the current description is somewhat inappropriate. I understand what the authors want to say, but it is better if the description is improved.*

Response: We agree with the reviewer that the previous description in Fig. 5 may be misleading. We rephrased it in a way that the bare Pd/Fe produces both types of products, while the Pd-C/Fe selectively catalyze C-O cleavage and semi-hydrogenation.

Actions taken: We have revised Fig. 5 accordingly as shown below.

Fig. 5. Schematic overview of the working mechanisms for the selective hydrogenolysis of DPE and selective hydrogenation of acetylene over a catalyst coordinated with alkenyl carbon.

Comment 15: *I believe this study is highly valuable. However, I feel the title “by mimicking homogeneous analogues” is inappropriate, because the active site is not Pd inside of metallacycle but far from the ligands. Thus, I recommend to change the title; for example, “Tuning hydrogenation chemistry of Pd-based heterogeneous catalysts by introducing homogeneous analogue ligands”.*

Response and action taken: We agree that the current title cannot accurately elucidate the mechanism in the selective catalysis. We have changed the title to “Tuning hydrogenation chemistry of Pd-based heterogeneous catalysts by introducing homogeneous-like ligands.”

Reviewer #2 (Remarks to the Author):

Comment 1: *The paper includes a very extensive work with the use of numerous techniques, however, the results do not clearly evidence the higher electron-density of Pd in 5Pd-C/Fe-C and its origin from carbonaceous compounds. Moreover, the benefit of a high electron-density of Pd for hydrogenation reactions in general cannot be claimed as new because it is already reported for other hydrogenation reactions. The following are the main points which need revision:*

Response: We express our gratitude to the reviewer for his/her valuable comments and helpful suggestions. In response to these comments, we have included additional analyses and experiments in the revised manuscript to further support our findings on the higher electron density and the role of carbonaceous compounds. Regarding the benefits of the high electron-density of Pd for hydrogenation, we acknowledge that this is not a new concept, as it has been recognized in numerous works. However, we believe it is important to reiterate its significance in our study, even though it is not directly associated with the major novelties of our work.

In the previous publications related to this field, various types of carbon (carbide, graphitic, chain-like, etc.) and molecular structures have been reported. Unfortunately, the carbon species in majority of those works co-exist and were not isolated, making the conclusions ambiguous and filled with all

kinds of speculations. In the present work, as reviewer1 also recognized, by controlling an appropriate pretreatment (i.e., the concentration of carbon precursor gas (i.e., CO) in H₂ and a carefully following H₂ treatment), we have selectively excluded different carbon spectators and identified the nature of active sites (e.g., metallacycle structure, coordination and working mechanism) with a complimentary of molecular-level characterizations including NMR of Pd-C labelled by ¹³C, NEXAFS, in situ-XPS and DFT modelling, etc., which, to our best knowledge, have never been reported before.

Comment 2: *Firstly, the use of alkenyl-type ligands to induce high electron-density to Pd for C-O bond cleavage do not seem to be justified because though selectivity is greatly increased, conversion is drastically reduced and therefore the yield to C-O bond cleavage products do not improve (Figure S.5).*

Response: We respectively disagree with the reviewer on this argument. As discussed at the beginning of this manuscript, Pd serves two functions: the first is to facilitate the C-O cleavage by enhancing the activation of H₂, and the second is to catalyze the unselective saturation of the aromatic ring. Therefore, two parallel reactions contribute to the total conversion over the Pd/Fe-based catalyst. The alkenyl-type ligands coordinating Pd selectively inhibited the direct hydrogenation of aromatic ring without influencing direct C-O cleavage. Therefore, the reduced conversion is expected since the direct ring saturation side reaction is inhibited. However, the yield to C-O bond cleavage was barely affected in our work as opposed to the trade-off between selectivity and activity of desirable reaction in the previous publications (Nat. Commun. 2013, 4, 2448; Nat. Catal. 2020, 3, 446-453). These details have been discussed in the Discussion section on Page 19 and 20:

“..... two common properties of noble metals during hydrotreating reactions are activation of H₂ and hydrogenation of unsaturated bonds. Our study presents an efficient approach using homogeneous-like surface Pd-alkene metallacycle on a heterogeneous Pd catalyst to tailor the electronic interaction between the sp² carbon in reactants and surface Pd so that the deep hydrogenation is inhibited. Moreover, the coordinative alkenyl ligand shows limited influence on activation of H₂ on Pd and C-O bond cleavage on Fe. This enables the selective utilization of the Pd character with inhibited negative effect in the reaction.”

“The homogeneous-like alkenyl ligand donates electrons to Pd to tailor the electronic property that elongates the distance and weakens interaction between the Pd and sp²-hybridized carbon in the reactants, leading to the inhibition of deep saturation (Surface Chemistry #2 in Fig. 5). Moreover, this unique type of ligand shows no inhibition to the rate of desired reaction pathway, that is distinct from the reported alkanethiolate-modified Pd on which a dramatic drop of production rate of desired products was observed⁵².”

Comment 3: *If Pd has a higher electron density in 5Pd-C/Fe-C catalysts, a displacement of normalized absorption with energy should be observed by Pd K edge XANES with respect the foil and the other catalysts. This is not clearly observed in Figure S.12. To better find out if this displacement exists would be necessary to represent XANES first derivative as a function of energy.*

Response: We thank the reviewer for this suggestion. It should be noted that the XANES is a technique measuring the bulk, given the small amount of surface carbon on the Pd, the displacement is expected to be small. The first derivative of XANES spectra of Pd (Supplementary Fig. S21) were used to locate the energy of absorption edge (E₀). The E₀ of 5Pd-C/Fe-C (24349.6 eV) is lower than those of Pd foil

(24350.0 eV), 5Pd/Fe (24349.9 eV) and 5Pd/Fe-C (24349.8 eV), indicating the Pd species over 5Pd-C/Fe-C is more electron-rich (J. Am. Chem. Soc. 2017, 139, 7294–7301). This is consistent with the XPS results (Fig. 3c).

Actions taken: We have added the first derivative of XANES spectra of Pd in Supplementary Information Fig. S21 and the corresponding discussion on Page 12 in the main manuscript.

“This is consistent with the first derivative of XANES (Supplementary S21) showing that the energy of absorption edge (E_0) of 5Pd-C/Fe-C (24349.6 eV) is slightly lower than those of Pd foil (24350.0 eV), 5Pd/Fe (24349.9 eV) and 5Pd/Fe-C (24349.8 eV), which indicates the Pd species over 5Pd-C/Fe-C is more electron-rich³⁶.”

Supplementary Fig. S21 The first derivative of XANES of catalysts and Pd foil at Pd K-edge.

Comment 4: Pd dispersion values and catalysts surface area are not found in the manuscript. They should be reported. In view of XRD results (Fig. S14), it seems that 5Pd-C/Fe-C shows the smallest Pd particles, it is reported in the bibliography that smallest Pd particles are prone to present a higher electron-density. To rule out the effect of metal particle sizes in the electronic properties of Pd, correlation of all these properties and catalytic ones must be analyzed in detail.

Response: The dispersion of Pd was analyzed by chemisorption of CO pulse (Supplementary Fig. S3), the details of which were described in the Supplementary Information. To address the reviewer’s comment, surface area and number of exposed Pd and Fe (used to calculate the TOFs) have been added as Table 1 in Supplementary.

We agree with the reviewer that particle size has been widely reported to strongly influence charge transfer and the electronic properties of supported metals. In this study, the combined CO pulse chemisorption (Fig. S3), XAS (Fig. S12, 13 and Table S2) and TEM analyses show that the particle size of three 5-wt.-%-Pd catalysts are similar. Therefore, it is inferred that the effect of Pd particle size has limited contribution to the distinct performances of 5Pd/Fe, 5Pd/Fe-C and 5Pd-C/Fe-C.

Actions taken: We have added a table summarizing the surface area, amounts of exposed Pd and Fe,

as well as the dispersion data in the Supplementary Information. It is referred on Page 6 in the main manuscript.

“..... the calculated turnover frequency (TOF) of C-O bond cleavage and direct hydrogenation were based on exposed Fe and Pd, respectively (details are shown in Supplementary Method, Fig. S3 and Table S1).”

Supplementary Table S1 Structural properties of the catalysts.

	Surface area (m ² /g)	Number of exposed Pd (mmol/g) ^a	Number of exposed Fe (mmol/g)	Dispersion of Pd (%) ^a	Particle size of Pd (nm) ^a	Particle size of Pd (nm) ^b
Fe	8.2	\	0.19	\	\	\
0.1Pd/Fe	8.1	0.011	0.18	81.7	1.4	-
1Pd/Fe	8.0	0.027	0.15	20.2	5.5	-
5Pd/Fe	9.2	0.055	0.16	8.8	12.7	10.8
5Pd/Fe-C	9.2	0.052	0.16	8.3	13.4	11.2
5Pd-C/Fe-C	9.4	0.048	0.17	7.7	14.5	10.7

a) Determined by CO pulse chemisorption.

b) Determined by TEM.

Comment 5: Carry on with the effect of Pd particle size, is found in the literature that small Pd particle sizes favour selectivity to ethylene in other hydrogenation reactions, therefore the influence of Pd particle size in the target reactions of the manuscript should be studied in detail. In that sense, as stated above, metal particle sizes should be clearly reported for all the catalysts, and it would be worthy to determine metal particle sizes from TEM and/or DRX measurements.

Response: We agree with the reviewer that small Pd clusters or single atoms favor the semi-hydrogenation of acetylene to ethylene. In fact, as mentioned in the response to the comment above, the CO chemisorption and XAS both show a similar particle size of Pd over 5Pd/Fe, 5Pd/Fe-C and 5Pd-C/Fe-C. Following the reviewer’s suggestion, we performed additional HRTEM analysis that also substantiates that these three catalysts have similar Pd particle size. Therefore, the difference of catalytic performances should be ascribed to the carbon ligand as we claimed.

It should be noted that the 5Pd-C/Fe-C has a larger deviation between particle sizes of Pd determined by CO pulse chemisorption and TEM, than the other two catalysts. This may be due to the ligand on Pd occupies a few sites so that the particle size is overestimated by using CO chemisorption method.

Actions taken: We added TEM analysis of 5Pd/Fe, 5Pd/Fe-C and 5Pd-C/Fe-C in Fig. S14 in Supporting Information. The corresponding discussion and experimental/analytical methods were added on Page 8 of main manuscript and Page 2-3 of Supporting Information.

In the main manuscript, “The particle sizes of Pd over 5Pd/Fe, 5Pd/Fe-C and 5Pd-C/Fe-C were first determined via the CO chemisorption results (12-15 nm, Supplementary Table S1). The

comparable sizes were further substantiated by a statistical analysis of microscopic images of these catalysts (Supplementary Fig. S14 and Table S1). It should be noted that 5Pd-C/Fe-C has a larger deviation between particle sizes of Pd determined by CO pulse chemisorption and TEM, than the other two catalysts. This may be due to that the ligands on Pd occupy certain sites so that the particle size is overestimated by CO chemisorption method. Overall, these results suggest the Pd and Fe in the three catalysts share identical structures, i.e., metallic Pd nanoparticles of similar size supported on bulk metallic Fe.”

In Supporting Information, “The dispersion was determined by assuming a chemisorption stoichiometry of one CO molecule per surface Pd atom¹. The mean size of Pd nanoparticle (d_m) was estimated with an assumption of hemispherical morphology using the reported equation²: d_m (nm) = $10^{21} \times (6 \times M \times \rho_{\text{surface site}}) / (D \times \rho_{\text{Pd}} \times N)$, where M is the atomic weight of Pd (106.4 g/mol), $\rho_{\text{surface site}}$ is the surface site density of Pd (12.7 atoms/nm²)³, D is dispersion, ρ_{Pd} is the metal density (12.0 g/cm³) and N as the Avogadro constant, giving d_m (nm) = 1.12/ D .”

“The mean particle diameter was calculated as previously reported⁴: $d_m = \sqrt{(\sum n_i \times d_i^2) / \sum n_i}$ ”

Supplementary Fig. S14 TEM images and Pd particle size distributions of 5Pd/Fe (a), 5Pd/Fe-C (b) and 5Pd-C/Fe-C (c). The analyses are based on 80–120 particles per sample.

Comment 6: It is reported in the manuscript that in STEM images carbon compounds are not associated to Pd in Pd/Fe-C (Fig 2.c), but reviewer do not agree with this assertion because although with less intensity, a higher intensity of C patterns in the areas of brightness of Pd is observed.

Response: In fact, we mentioned that “carbon is detected on both Pd and Fe domains of 5Pd/Fe-C” in the discussion of STEM images. As for the schematic illustration of carbon locations on Pd and Fe (Fig. 2e in the revised version), we agree that the previous discussion about the three catalysts is not

appropriate. To avoid any possible confusion, we revised the corresponding discussion and rearranged the layout of Fig. 2 to cite the figures more precisely for a particular discussion.

Actions taken: We changed the layout of the previous Fig. 2b-d to current Fig. 2b-e. The corresponding discussion on Page 10 was revised.

Fig. 2. Microscopic analysis of Pd/Fe, Pd/Fe-C and Pd-C/Fe-C. **a**, HRTEM images of samples, the left and right ones show the regions in white squares; **b-d**, STEM images and EDS elemental mapping of Pd, C, Fe over 5Pd/Fe (**b**), 5Pd/Fe-C (**c**) and 5Pd-C/Fe-C (**d**), Brown: Pd, Green: C, Navy: Fe; **e**, schematic illustration of carbon locations on Pd and Fe. All samples were pretreated with 10% H_2/Ar at 250 °C for 1 h to simulate the catalysts after the hydrotreating reaction.

“Based on the above results, structures of the three catalysts are proposed and schematically presented in Fig. 2e. Compared with the clean 5Pd/Fe and the 5Pd/Fe-C that show C on both Pd and Fe region, the Pd in 5Pd-C/Fe-C is covered with a significantly higher amount of C species.”

Comment 7: *Is there any alloy Pd-Fe formation? It is known that the addition of a second metal, and particularly Fe, can modify electronic properties of Pd. Moreover, the addition of a carbon treatment could modify the reduction atmosphere and modify the interaction of both metals modifying the effect of Fe on Pd particles instead being the carbonaceous species properly those modifying electronic properties of Pd. ¿Why can be this effect discarded?*

Response: Thank you for this interesting question. In our previous work (Journal of Catalysis 2013, 306, 47–57), we reported that Pd-Fe alloy is indeed identified by HRTEM on the PdFe bimetallic catalyst. In this work, our EXAFS results also revealed alloy formation on 5Pd/Fe, 5Pd/Fe-C, and 5Pd-C/Fe-C catalysts (Supplementary Fig. S13 and Table S2). It also showed that 0.4vol.% CO/H_2 treatment does not seem to change the CN of Pd-Fe. More importantly, no selective chemistry was observed on 5Pd/Fe, which seemingly excluded the Pd-Fe alloy effect. In addition, the Fe-free catalysts (i.e., CNT-supported ones) also show improved selective hydrogenation chemistry as the Fe-supported ones after the similar treatment with 0.4vol.% CO/H_2 (Supplementary Fig. S22 and S23). Those results suggest Fe-Pd alloy or Fe modified Pd should not play a major role in regulating the catalytic performance, though further evidence may be helpful to unambiguously confirm whether the treatment using

0.4vol%CO/H₂ will change the Pd-Fe interaction or not. Regardless, our results strongly suggest that the metallacycle type carbon ligand and its interaction with Pd should play the key roles in this selective chemistry.

Actions taken: The possibility that the reviewer mentioned has been discussed on Page 12 in the revised manuscript.

“Besides the direct interaction between Pd and carbon species, one possible mechanism of electronic modification of Pd is that carbon species affects the Pd-Fe interaction. However, the EXAFS results showed no obvious change of Pd-Fe coordination number by a 0.4 vol.% CO/H₂ treatment (Supplementary Fig. S13 and Table S2). More importantly, no selective chemistry was observed on 5Pd/Fe, suggesting the Pd-Fe alloy effect should be excluded. In addition, the Fe-free catalysts (i.e., CNT-supported ones) also show improved selective hydrogenation chemistry as the Fe-supported ones after the similar treatment with 0.4 vol.% CO/H₂ (Supplementary Fig. S22 and S23). Those results suggest that Fe-Pd alloy or Fe modified Pd should not play a major role in regulating the catalytic performance.”

Comment 8: *To better rule out the effect of Fe in the electronic properties of Pd a Pd-Fe/CNT catalysts should be also studied in addition to Pd/CNT and Pd-C/CNT.*

Response: As discussed in our response to the comment above, our current results have ruled out Fe as the main contributor to the selective catalysis. Particularly, the Pd/CNT still shows enhanced selective hydrogenation chemistry after the pretreatment with 0.4vol%CO/H₂. It is clear that only when the Pd is modified with C (i.e., 5Pd-C/Fe-C and Pd-C/CNT), can the selectivity be significantly enhanced. In addition, we cannot guarantee the homogeneity of metal sites, i.e., Pd-Fe, Pd and Fe will likely co-exist on surface, that may complicate the investigation. Therefore, we believe that the investigation of a Pd-Fe/CNT would not provide essential understanding about the role of Fe, given that we have ruled out Fe as the main contributor to the selective catalysis based on the current results.

Comment 9: *¿Why does products distribution change in such a significant extent for 5Pd-C/Fe-C?*

Response: Pd domain is responsible for the deep hydrogenation. The experimental and theoretical results show that the alkenyl ligand on Pd donate electrons to Pd, creating an electron-rich environment. Such a modification weakens the electronic interaction between Pd and sp²-hybridized carbon of the reactants/products to control the hydrogenation chemistry. As a result, the direct hydrogenation of DPE and deep hydrogenation of ethylene is significantly inhibited, leading to the dramatic change of products distribution (Fig. 1c and 1d).

Comment 10: *As NMR has been performed without Fe in the catalysts, the findings are questionable as the catalyst change.*

Response: The catalysts in NMR are clean Pd/CNT and Pd-¹³C/CNT. Though Pd is not supported on Fe, the changes of catalytic performance for both C-O cleavage and acetylene semi-hydrogenation shown below are similar with the Pd-Fe-based catalysts (i.e., C coordination on Pd inhibits the deep hydrogenation). It suggests that these two batches of catalysts (i.e., CNT and Fe as supports) share similar structure of Pd domain. This has been carefully discussed on Page 12-13 in the main manuscript: “..... the magnetic Fe support was replaced by carbon nanotube since the tailored hydrogenation chemistry mainly occurs on Pd. Indeed, the selective hydrogenation chemistry of ¹³CO-treated Pd/CNT

was verified by the inhibition of ring saturation during DPE hydrotreating (Supplementary Fig. S22). Selective hydrogenation of acetylene to produce ethylene was also confirmed on the Pd-C/CNT catalysts (Supplementary Fig. S23).”

Supplementary Fig. S22 Catalytic performances of Pd/CNT and Pd-¹³C/CNT. The amount of Pd is 5 wt.%. Reaction condition: 0.005 g catalyst, 50 mL C₁₆H₃₄, 1.08 g DPE, 250 °C, 5.6 MPa H₂.

Supplementary Fig. S23 Performance of catalysts for hydrogenation of acetylene: conversion (a) and ethylene selectivity (b). Reaction condition: 0.012 g catalyst, 4% C₂H₂-8% H₂ balanced with N₂, GHSV=120,000 mL/(g*h), 150 °C, 0.1 MPa.

Comment 11: *The position of figure 2 within the text is not suitable.*

Response and action taken: We have moved Fig. 2 and its caption to the position where the corresponding discussion starts.

Comment 12: *The description of preparation of CNT supported Catalysts (lines 392-396 of page 20) is not clear. It is confusing which catalysts have been prepared.*

Response: We have revised the corresponding section to provide a more detailed description of the synthesis procedure for the CNT-supported catalysts.

Actions taken: The description of preparation of CNT supported catalysts on Page 21 has been revised as shown below.

“The impregnation process to synthesize bare Pd/CNT (i.e., 5 wt.% Pd supported on carbon nanotube) was the same as that of 5Pd/Fe except using carbon nanotube as support. The dried sample was then reduced with UHP H₂ (50 mL/min) at 350 °C for 2h. Pd-¹³C/CNT (i.e., Pd/CNT decorated by ¹³C labelled carbon) was prepared with same procedure as Pd/CNT except using 0.4 vol.% ¹³CO (99% abundancy, Cambridge Isotope Laboratories, Inc.) balanced by H₂ instead of UHP H₂ in the reduction process. The sample was then pretreated with 10 vol.% H₂/He (50 mL/min) at 250 °C for 0.5 h. Both catalysts were passivated using the same procedure as described above.”

Comment 13: *A comparison of activity and selectivity of Pd-C/CNT and 5Pd-C/Fe-C should be included.*

Response: To address the reviewer’s comment, we have included the comparison of acetylene hydrogenation performances on these two catalysts.

As for activity and selectivity of C-O cleavage, although Pd dissociate H₂, H was transferred to Fe which is the main active site for C-O cleavage in Pd/Fe catalyst. Therefore, Pd-C/CNT that contains no Fe site cannot be compared with 5Pd-C/Fe-C for a fair comparison, regarding the DPE conversion.

Actions taken: We have added comparison about acetylene hydrogenation in the Supplementary Information.

Supplementary Fig. S24 Performances of Pd-C/CNT (0.012 g), and 5Pd-C/Fe-C (0.03 g) for hydrogenation of acetylene: conversion (a) and ethylene selectivity (b). Reaction condition: 4% C₂H₂-8% H₂ balanced with N₂, GHSV=120,000 mL/(g*h), 150 °C, 0.1 MPa. The data was derived from Supplementary Fig. S10 and S23 at 60 min.

Considering the low density of CNT, the weight of Pd-C/CNT was lower than 5Pd-C/Fe-C to decrease the pressure drop. However, the same GHSV was maintained.

In the main manuscript (Page 13): "...A comparison of the catalytic performances of Pd-C/CNT and 5Pd-C/Fe-C for acetylene hydrogenation is shown in Supplementary Fig. S24."

REVIEWERS' COMMENTS

Reviewer #1 (Remarks to the Author):

The authors have appropriately answered Reviewers' comments. Thus, the revised version is basically worthy for publication. However, one question has arisen. The authors have provided a new list of number of exposed Pd and Fe, Pd dispersion, particle size, etc. in Supplementary Table S1. The number of exposed Pd was estimated from CO pulse adsorption, and the number of exposed Fe was estimated by subtracting the exposed Pd from BET surface area. I know CO also chemisorbs on Fe. Is the amount of CO molecules adsorbed on Fe negligible compared to that on Pd? Data of CO pulse adsorption on the Fe sample listed in Table S1 will give an answer to this question.

Reviewer #2 (Remarks to the Author):

I find that the revised version has attended the points raised and I recommend to publish the manuscript

Reviewer #1 (Remarks to the Author):

The authors have appropriately answered Reviewers' comments. Thus, the revised version is basically worthy for publication. However, one question has arisen. The authors have provided a new list of number of exposed Pd and Fe, Pd dispersion, particle size, etc. in Supplementary Table S1. The number of exposed Pd was estimated from CO pulse adsorption, and the number of exposed Fe was estimated by subtracting the exposed Pd from BET surface area. I know CO also chemisorbs on Fe. Is the amount of CO molecules adsorbed on Fe negligible compared to that on Pd? Data of CO pulse adsorption on the Fe sample listed in Table S1 will give an answer to this question.

Response: We thank the reviewer for the positive comment and the great question/suggestion. The CO pulse experiment was conducted at 30 °C where the amount of CO molecules adsorbed on Fe is negligible, as shown in Supplementary Fig. 3a. The calculated value of adsorbed CO on monometallic Fe is 6×10^{-4} mmol/g, around 1% of CO on a Pd containing 5Pd/Fe catalyst (550×10^{-4} mmol/g). We have added the data on Fe in the annotation of Supplementary Table 1.

Supplementary Fig. 3 Results of CO pulse experiments at 30 °C on monometallic Fe (a), 0.1Pd/Fe (b), 1Pd/Fe (c), 5Pd/Fe (d), 5Pd/Fe-C (e) and 5Pd-C/Fe-C (f). Prior to pulsing CO, the catalysts were pretreated with $10\text{H}_2/\text{Ar}$ at 250 °C for 1h to simulate the catalysts during the reaction. The signal of CO was recorded at $m/z=28$.

Revised Supplementary Table 1 (the revision parts are highlighted):

Supplementary Table 1 Structural properties of the catalysts.

	Surface area (m ² /g)	Number of exposed Pd (mmol/g) ^a	Number of exposed Fe (mmol/g)	Dispersion of Pd (%) ^a	Particle size of Pd (nm) ^a	Particle size of Pd (nm) ^b
Fe	8.2	\ ^c	0.19	\	\	\
0.1Pd/Fe	8.1	0.011	0.18	81.7	1.4	-
1Pd/Fe	8.0	0.027	0.15	20.2	5.5	-
5Pd/Fe	9.2	0.055	0.16	8.8	12.7	10.8
5Pd/Fe-C	9.2	0.052	0.16	8.3	13.4	11.2
5Pd-C/Fe-C	9.4	0.048	0.17	7.7	14.5	10.7

a) Determined by CO pulse chemisorption.

b) Determined by TEM.

c) The calculated value using the CO pulse chemisorption result is $\sim 6 \times 10^{-4}$ mmol/g, indicating that adsorption of CO on Fe is negligible for the calculation of numbers of exposed Fe and Pd (detailed procedure is shown in Supplementary Method).

Reviewer #2 (Remarks to the Author):

I find that the revised version has attended the points raised and I recommend to publish the manuscript.

Response: We thank the reviewers for the positive comment and the recommendation of publication.